

# High speciation in the cryptic *Pristimantis celator* clade (Anura: Strabomantidae) of the Mira river basin, Ecuador-Colombia

Mario H. Yánez-Muñoz[1], Juan P. Reyes-Puig[1,2,3], Carolina Reyes-Puig[1,3,4], Gabriela Lagla-Chimba[1], Christian Paucar-Veintimilla[1], Miguel A. Urgiles-Merchán[1] and Julio C. Carrión-Olmedo[1]

[1] Unidad de Investigación, Instituto Nacional de Biodiversidad (INABIO), Quito, Pichincha, Ecuador
[2] Red de Bosques Amenazados, Fundación Ecominga, Baños, Tungurahua, Ecuador
[3] Departamento de Ambiente, Fundación Oscar Efren Reyes, Baños, Tungurahua, Ecuador
[4] Instituto de Biodiversidad Tropical IBIOTROP, Museo de Zoología & Laboratorio de Zoología Terrestres, Colegio de Ciencias Ambientales, Universidad San Francisco de Quito (USFQ), Quito, Pichincha, Ecuador

Corresponding author
Mario H. Yánez-Muñoz,
mayamu@hotmail.com

## ABSTRACT

Over the past decade, research in the montane forests of the Mira River basin, spanning Ecuador and Colombia, has identified it as crucial for the adaptive radiation of flora and fauna, shaped by its complex geological and climatic history. This study focuses on the phylogenetic and systematic revision of a frog clade initially labeled as *Pristimantis verecundus*, revealing significant cryptic diversity. Through detailed analyses of type material and expanded molecular sampling, we found that the original description actually included specimens representing two additional species, which are described herein. In this work, we discovered and formally described four new species within montane forests at elevations from 1,600 to 2,300 meters. Genetic distances of 3.34% to 14% and clear morphological differences underscore the clade's hidden diversity. We propose renaming the group *Pristimantis celator* clade within *Pristimantis myersi* species group and subgenus *Trachyphrynus*, aligning with phylogenetic evidence and resolving taxonomic ambiguities using the oldest available name, *Pristimantis celator* (Lynch, 1976). This reclassification includes 14 species, seven formally described, and seven as candidates, distributed across northwestern Ecuador and southwestern Colombia, particularly in Mira and Esmeraldas River basins. The study highlights the Andean orogeny's role in species diversification within *Pristimantis celator* clade, with geographic barriers like Cerro Golondrinas influencing genetic isolation. Genetic divergences exceeding 3.34% indicate evolutionary isolation across these landscapes. Our findings provide insights into montane ecosystem speciation, emphasizing vicariance, niche adaptation, and altitudinal gradients in shaping biodiversity. A polytomy among three well-supported clades within *Pristimantis myersi* species group is noted due to incomplete genetic data, yet distinctiveness and evolutionary relationships are affirmed. Cryptic diversity within *Pristimantis celator* clade links to unique orogenic and climatic conditions, highlighting conservation needs. Lastly, we provide a redescription of *Pristimantis verecundus* and species identification key to aid future research and conservation in this biogeographically influential region.

# INTRODUCTION

The Tropical Andes represent one of the most diverse regions globally in terms of ecosystems, habitats, and endemic herpetofauna species. This diversity arises from the region's complex geography, geology, elevational, and climate patterns (*Duellman, 1979*; *Lynch & Duellman, 1997*; *Pinto-Sánchez et al., 2015*; *Mendoza et al., 2015*; *Vasconcelos et al., 2019*). In northwestern Ecuador, the Mira River drainage cuts through the western cordillera of the Andes, forming a deep canyon that connects lowland Chocoan rainforests with high-altitude ecosystems in a continuous line. This geographical feature promotes high levels of diversification and endemism, particularly within the Mira watershed and its mountain ranges. This pattern is evident across several groups of small vertebrates, including amphibians, reptiles, and rodents (*Yánez-Muñoz et al., 2018*; *Yánez-Muñoz et al., 2021a*; *Yánez-Muñoz et al., 2021b*; *Yánez-Muñoz et al., 2024*; *Brito et al., 2020*; *Reyes Puig et al. 2020a*; *Reyes Puig et al. 2020b*).

In addition to the broad patterns of diversity across the region, a specific group of mountains located west of Cerro Golondrinas and east of the Awa Indigenous Territories stands out. This area is among the richest in orchids and birds within the Tropical Andes and Chocó regions (*Santander, Freile & Loor-Vela, 2009*). Notably, it encompasses the Dracula Reserve, which was established to protect key endangered forests and serve as a type locality for newly described batrachians and lizard species (*Yánez-Muñoz et al., 2018*; *Yánez-Muñoz et al., 2021a*; *Yánez-Muñoz et al., 2021b*; *Reyes-Puig et al., 2020a*; *Reyes-Puig et al., 2020b*).

The direct-developing terrestrial frog genus *Pristimantis* Jiménez de la Espada 1890, thrives in the northern Andes of South America, where its unique environment boosts rapid lineage divergence, which continues to increase in this region (*Pinto-Sánchez et al., 2015*; *Mendoza et al., 2015*; *Vasconcelos et al., 2019*; *Yánez-Muñoz et al., 2020*; *Reyes-Puig & Mancero, 2022*). As systematic and phylogenetic studies of the genus deepen, challenges in understanding its diversity and ancestor-descendant relationships are revealed. The *Pristimantis myersi* species group is particularly diverse and cryptic, with high genetic diversity and numerous new nominal species awaiting description (*Franco-Mena et al., 2023*).

Closely related to this group, *Pristimantis verecundus* (*Lynch & Burrowes, 1990*) was initially described from a type series with varying coloration patterns, but little is known about the variation in live specimens; a detailed examination of the type material of *Pristimantis verecundus* from Reserva La Planada, revealed contrasting differences between specimens, suggesting that cryptic taxa are blended within the type series. This species has recently shown a polytomy, suggesting the increase of new taxa, and is currently grouped in the *P. verecundus* clade (*Franco-Mena et al., 2023*). Over the past decade, we have conducted systematic expeditions in the extreme northwestern region of Ecuador, which allowed us to collect, examine, and obtain fresh genetic material to better understand this group. In the present work, we increase the phylogenetic sampling of the clade, revealing four new species described here and three additional unnamed species.

## MATERIALS & METHODS

### Ethics statement

This research was follow standards of the Ministry of Environment, Water, and Ecological Transition of Ecuador MAATE, under permits granted for research and access to genetic resources: MAE-DNB-CM-2016-0045, N° MAE-DNB-CM-2019-0120, and MAATE-ARSFC-2023-3346 issued. We follow ethic statements suggested by *Beaupre et al. (2004)* established by the American Society of Ichthyologists and Herpetologists, the Herpetologists' League, and the Society for the Study of Amphibians and Reptiles.

Examined specimens were sourced from the herpetology repositories (Appendix 1): Instituto Nacional de Biodiversidad INABIO, Quito, Ecuador (DHMECN); Instituto Humboldt, Villa de Leyva, Colombia (IAvH, acronym previously known as IND-AN); and Museo de Zoología Universidad San Francisco de Quito, Quito, Ecuador (ZSFQ). Museum acronyms follow *Frost (2023)*.

### Taxon sampling

For the taxonomic descriptions, we used a combination of several lines of evidence, including external morphological characters, linear morphometric variations, genetic divergence, and geographic distribution. We selected a comprehensive line of evidence to describe the new species, including representative variation in the type series observed both in life and in preservation. We examined high-resolution photographs of the type material of *Pristimantis verecundus* (*Lynch & Burrowes, 1990*) provided by the IAvH (Figs. S1–S2). Similar approaches have been used by us to recognize and identify cryptic amphibian and reptile complexes in the Mira river basin landscape (*Yánez-Muñoz et al., 2018*; *Yánez-Muñoz et al., 2021a*; *Yánez-Muñoz et al., 2021b*; *Reyes-Puig et al., 2020a*; *Reyes-Puig et al., 2020b*).

Names of new species reported herein will represent a published work according to the International Commission on Zoological Nomenclature (ICZN), are effectively published under that Code from the electronic edition alone, the nomenclatural acts it contains have been recorded in ZooBank, the online registration system for the ICZN. The ZooBank LSIDs (Life Science Identifiers) can be found by appending the LSID to the prefix http://zoobank.org/. The LSID for this publication is: urn: lsid:zoobank.org:pub:0E31ADC7-D760-427D-A7F6-5E6ADB7A21B5.

### Field work

Herpetological surveys were steered using visual encounter surveys (*Rueda, Castro & Cortez, 2006*). Field work was conducted in the mountains and tributary drainages of the Mira watershed, during expeditions of Instituto Nacional de Biodiversidad (INABIO) and the Ecominga Foundation, in 10 locations sampled between the years 2015 to 2023, shown in Table 1. Like other previous research (*Reyes-Puig et al., 2020a*; *Reyes-Puig et al., 2020b*; *Yánez-Muñoz et al., 2021a*; *Yánez-Muñoz et al., 2021b*), the specimens collected were photographed alive, euthanized with benzocaine, a sample of muscle tissue was extracted and preserved in 95% ethanol; the individuals were fixed in 10% formalin, and preserved in 75% ethanol, as referred in (*Rueda, Castro & Cortez, 2006*).

**Table 1  List of localities sampled in the Mira River Basin in Ecuador for this research.**

| No | Locality | Collectors | Geographic coordinates | Elevation in meters above sea level | Collection date |
|---|---|---|---|---|---|
| 1 | Reserva Dracula, Sector Cerro Oscuro | Mario H. Yánez-Muñoz, Juan P. Reyes-Puig & Jorge Brito M. | 0.898426 -78.207942 | 1,600 | From: 2015-07-18 to: 2015-07-22 |
| 2 | Reserva Dracula, Sector El Pailón | Mario H. Yánez-Muñoz & Juan P. Reyes-Puig | 1.0125630 -78.24620 | 1,450 | From: 2017-11-08 to: 2017-11-12 From: 2018-04-23 to: 2018-04-28 |
| 3 | San Jacinto de Chinambí | Mario H. Yánez-Muñoz, Mateo Vega Yánez & Daniel Padilla | 0.860927 -78.272364 | 2,600 | From: 2019-06-8 to: 2019-06-15 |
| 4 | Sector La Esperanza | Mario H. Yánez-Muñoz, Juan P. Reyes-Puig, Evelyn Gabriela Lagla Chimba, & Miguel Urgiles | 0.937383 -78.2425 | 1,670 | From: 2021-03-24 To 2021-03-30 |
| 5 | Bosque Protector Cerro Golondrinas, Sector Cañas el Pailón | Miguel Andrés Urgilés Merchán, Christian Paucar Veintimilla | 0.82377 -78.09519 | 2,510 | From: 2023-07-25 To: 2023-08-01 |
| 6 | Reserva Dracula, Base del Cerro Golondrinas | Juan P. Reyes-Puig | 0.85784 -78.19729 | 2,093 | From: 2023-06-18 To: 2023-07-07 |
| 7 | Reserva Dracula, Sector Cerro Negro | Ross Maynard | 0.87102 -78.2069 | 2,221 | From: 2021-08-15 To: 2023-07-08 |
| 8 | Reserva Dracula, Base del Cerro Golondrinas | Julio C. Carrión, Pearson McGovern, Callie Broaddus | 0.88413 -78.2077 | 1,962 | From: 2021-08-03 to: 2021-08-12 |
| 9 | Reserva Dracula, Base del Cerro Golondrinas | Evelyn Gabriela Lagla Chimba & Julio C. Carrión | 0.86638 -78.202444 | 2,277 | From: 2022-08-03 to: 2022-08-12 |
| 10 | Reserva Dracula, Sector Los Olivos (Bloque 20) | Mario H. Yánez-Muñoz, Miguel Andrés Urgilés Merchán, Christian Paucar Veintimilla, Carlos Ríos | 0.83645 -78.24127 | 2,335 | From: 2023-12-05 to: 2023-12-14 |

## Morphological data

The taxonomic terminology follows the proposal of *Duellman & Lehr (2009)*. Morphometric measurements were taken with an electronic caliper (accuracy ± 0.01 mm, rounded to 0.1 mm). The following morphological measurements were taken following similar descriptions (*Duellman & Lehr, 2009*; *Reyes-Puig et al., 2020b*): (1) snout–vent length (SVL) = distance from snout tip to posterior margin of vent; (2) head width (HW) = greatest width of head measured at level of jaw articulation; (3) head length (HL) = from posterior margin of lower jaw to tip of snout; (4) horizontal eye diameter (ED) = distance between anterior and posterior borders of eye; (5) interorbital distance (IOD) = the breadth of the braincase between the orbits; (6) eye–nostril distance (EN) = distance from posterior margin of nostril to anterior margin of eye; (7) tympanic length (TD) = horizontal distance between external anterior and posterior margins of tympanic annulus; (8) internarinal distance (IND) = from the external border of nostrils; (9) tibia length (TL) = length of flexed leg from knee to heel; (10) upper eyelid width (EW) = perpendicular distance of the upper eyelid; (11) foot length (FoL) = distance from the proximal edge of the medial metatarsal tubercle to the tip of the fourth toe; (12) hand length (HaL) =

distance from proximal edge of palmar tubercle to tip of Finger III; (13) disc width of finger III (F3D) = measured across widest part of finger disc III; (14) disc width of toe IV (T4D) = measured across widest part of toe disc IV. Fingers are numbered pre-axially to post-axially from I–IV. Comparative lengths of toes III and V were determined, both compared to toe IV; lengths of toes I and II were estimated compared to each other. Sex, maturity of specimens, and reproductive condition were delimited by the identification of vocal slits, size, and through direct observation of gonads by dorsolateral sectioning.

Sexual dimorphism was analyzed descriptively by visualizing the variation in body size (SVL) between sexes using box plots (Fig. S3). The analysis was conducted using the statistical package Past® (*Hammer, Harper & Ryan, 2001*).

According to *Guayasamin et al. (2015)*, to minimize phenotypic plasticity within the clade, only diagnostic characters documented within 12 h of collection were considered. Color in life was determined based on photographs taken in the field from original collectors. In the descriptive section of living and preserved colorations of the type series of the new species, the colors are accompanied by the catalog number associated with the *Köhler (2012)* standard in parentheses.

## DNA extraction, amplification, sequencing, and bioinformatics

DNA extraction and PCR amplification followed the methodology described in *Reyes-Puig et al. (2024)*. Partial mitogenomes were amplified, flanking 12S rRNA to ND1 region. We targeted two overlapping >2,000 bp mitochondrial fragments using a mid-range PCR to amplify multiple genes at once. The first fragment flanks from 12S to 16S using 12sL4E (TACACATGCAAGTYTCCGC) with 16H36E (AAGCTCCAWAGGGTCTTCTCGTC) (*Heinicke, Duellman & Hedges, 2007*), with the following thermocycler protocol: 5 min @ 95 °C, then 35 cycles of: 45 secs at 95 °C, 35 secs at 50 °C, and 2 min at 72 °C, then 5 min at 72 °C. The second fragment flanks 16S-ND1 using 16L19 (AATACCTAACGAACTTAGCGATAGCTGGTT) with t-Met-frog (TTGGGGTATGGGCCCAAAAGCT) (*Wiens et al., 2005*; *Moen & Wiens, 2009*), with the following thermocycler protocol: initial denaturation 5 min @ 95 °C, 30 secs at 95 °C, 30 secs at 57 °C and 4 min at 72 °C for 35 cycles, with a final extension time of 5 min at 72 °C.

Chosen molecular markers (12S rRNA, 16S rRNA, and ND1) are widely used in amphibian phylogenetics (*Hay et al., 1995*; *Chan et al., 2022*; *Portik et al., 2023*).

Sequencing run was performed on a minION mk1c using Flongle Flow Cells R10.4.1 and Rapid Barcoding Kit 96 V14 (SQK-RBK114.96) following manufacturer protocols. Raw reads were high-accuracy (HAC) basecalled and demultiplexed with Dorado 7.3.11.

Consensus sequences were generated with NGSpeciesID (*Sahlin, Lim & Prost, 2021*) using above Q12 fastq reads. Consensus sequences expected divergence from Sanger between 0.00–0.04% (*Vasiljevic et al., 2021*), too small to be consequential to robust phylogenetic analysis. FASTQ files management, consensus generation, and subsequent FASTA files management, renaming, and concatenation were automated using a custom Python script available in Zenodo by *Carrión-Olmedo (2024)*.

DNA extraction, PCR amplification, nanopore sequencing, and bioinformatic analysis were performed at the Nucleic Acid Sequencing Laboratory of the Instituto Nacional de Biodiversidad (INABIO) in Quito, Ecuador. 29 consensus sequences were generated (20 of 12S, 29 of 16S, and 18 of ND1) of 29 individuals. Generated sequences were imported and interpreted using Mesquite (*Maddison & Maddison, 2018*).

The newly generated sequences are available in GenBank (Table S1) as suggested by *Chakrabarty et al. (2013)*. We expanded our study group to include species shown to be closely related to *Pristimantis verecundus* based on *Franco-Mena et al. (2023)* and an unpublished phylogeny of *Pristimantis* obtained by Julio C. Carrión-Olmedo as part of a large-scale review of northern *Pristimantis*.

Due to genetic data availability, we first constructed a 16S matrix to infer the phylogenetic position among congeners based on the phylogeny of *Franco-Mena et al. (2023)*. To infer a more robust phylogenetic placements of new lineages within the scope of this group we constructed a partitioned matrix with the partial mitogenomes. The partitions were as follows: 12S rRNA, tRNA-Val, 16S rRNA, tRNA-Leu, and codon positions in ND1. Matrices were aligned using default parameters in MAFFT (*Katoh & Standley, 2013*) and visually inspected for unambiguous alignment errors.

Substitution models and maximum likelihood tree inference were performed using IQ-TREE (*Trifinopoulos et al., 2016*) under default settings. Branch support was evaluated using 2,000 ultrafast bootstrapping and SH-aLRT tests with 1,000 replicates (*Guindon et al., 2010*). Support values mentioned herein follows this format: SH-aLRT support (%)/ultrafast bootstrap support (%). Uncorrected p-distances were calculated with an 800 bp-long fragment of 16S rRNA using MEGA 11 (*Tamura, Stecher & Kumar, 2021*).

## RESULTS

### Phylogenetic relationships and genetic distances (Fig. 1)

The 16S exploratory matrix was 1,653 bp long for 330 individuals and the partial mitogenomes matrix was 3,958 bp long for 99 individuals. IQTree evaluated the best substitution model for seven partitions as follows: TIM2+F+G4 for 12S rRNA, TIM2+F+G4 for tRNA-Val, TIM2+F+I+G4 for 16S rRNA, TIM2+F+G4 for tRNA-Leu, TIM2e+G4 for ND1 first position, TPM3+F+G4 for ND1 second position, TIM+F+G4 for ND1 third position.

The positioning of the study clade and species in the genus *Pristimantis* was highly supported (Fig. 1A). We expand the phylogenetic diversity knowledge and the number of species on the *Pristimantis verecundus* clade *sensu Franco-Mena et al. (2023)*, from 10 to 14 species (Fig. 1A). Like *Franco-Mena et al. (2023)*, we also identified the unresolved evolutionary relationships among the *Pristimantis myersi* species group (Fig. 1B.). Three well-supported clades formed a polytomy, the first clade (90.7/100 branch supports) is composed by *Pristimantis celator + P. mutabilis + P. verecundus* and several candidate species, the second clade (57.7/93 branch supports) is composed of *P. jubatus + P.* c.sp. 1, and the last clade is the *P. myersi* clade (79/96 branch supports).

An undescribed species from the clade of *Pristimantis verecundus* reported as *P.* sp. 3 from Via Ibarra-San Lorenzo by *Franco-Mena et al. (2023)* is here described as *P. praemortuus*

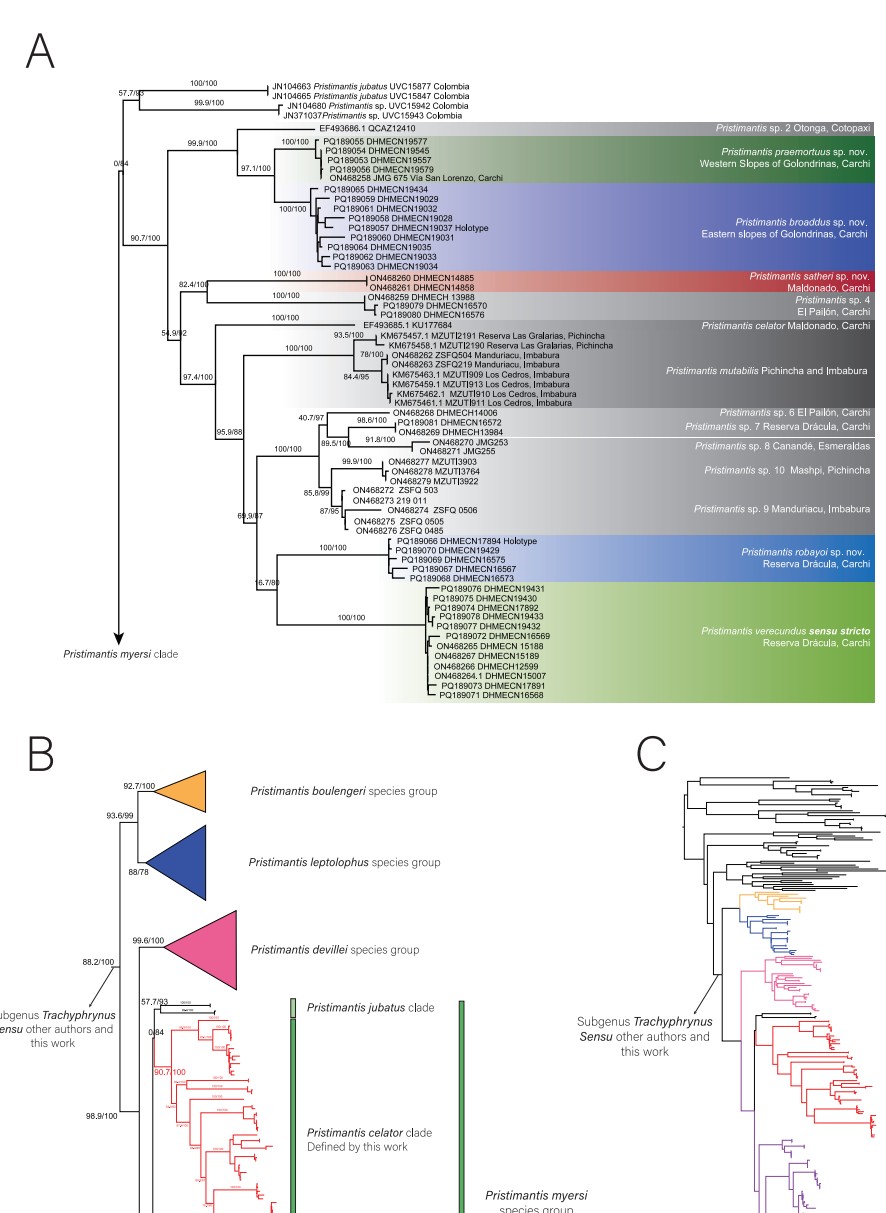

**Figure 1  Phylogeny of *Pristimantis celator* clade.** (A) Maximum Likelihood clade credibility tree obtained from 3,955 bp long for 63 individuals; (B) Relationship of *Pristimantis celator* clade to *P. myersi* species group and its sister clades; (C) Phylogenetic position of the *Pristimantis celator* clade and sister clades within the genus *Pristimantis,* from 330 indiviuals.

sp. nov. and it is sister species of *P. broaddus* sp. nov. (97.1/100 branch supports). Similarly, *Pristimantis* sp. 5 reported by *Franco-Mena et al. (2023)* is here described as *P. satheri* sp. nov. Additionally, as we increased the sampling, a new unreported lineage was included within the *Pristimantis verecundus* clade, here we describe it as *P. robayoi* sp.nov., a closely related species to *P. verecundus* (*Lynch & Burrowes, 1990*).

*Franco-Mena et al. (2023)* in nominating the *Pristimantis verecundus* clade, did not consider taxonomically that *Pristimantis celator* (*Lynch, 1976*) is the oldest available name for the clade. We define the *Pristimantis verecundus* clade (*sensu Franco-Mena et al., 2023*) nominally as the *Pristimantis celator* clade. We consider that despite the polytomy confirmed in this and other research (*Franco-Mena et al., 2023*), the *Pristimantis jubatus* clade and *Pristimantis celator* clade should be part of the *Pristimantis myersi* species group, into the subgenus *Trachyphrynus* (*Hedges, Duellman & Heinicke, 2008*; *Rivera-Correa & Daza, 2016*; *González-Durán et al., 2017*; *Jetz & Pyron, 2018*; *Bejarano-Muñoz et al., 2022*).

*Pristimantis celator* clade (Fig. 1A.) is composed of two subclades. The subclade A has high support (99.9/100) for grouping one candidate species (*P.* sp. 2) and two new sister species described herein (*P. broaddus* sp. nov. + *P. praemortuus* sp. nov.). The subclade B has a medium support (54.9/92) and contains a subclade of moderately high support (82.4/100) formed by *Pristimantis satheri* sp. nov. + one candidate species (*P.* sp. 4.). Which is a sister of a high support subclade (97.4/100) comprised of *Pristimantis celator*, *P. mutabilis*, five new candidate species (*P.* sp. 6, *P.* sp. 7., *P.* sp. 8., *P.* sp. 9., and *P.* sp. 10.), and *Pristimantis verecundus* with its closely related species *Pristimantis robayoi* sp. nov.

## Systematic accounts
### *Pristimantis (Trachyphrynus) celator* clade

**Definition.** Small-sized frogs with proportionally long limbs; ecotypes slender-bodied terrestrial and bush frogs; SVL in males ranges from 12.46 mm in males of *Pristimantis praemortuus* sp.nov. to 29.5 mm in females *P. robayoi* sp. nov. Dorsolateral dermal or glandular folds present (except in *Pristimantis celator*). Head width 33.76–39.6% of SVL. The tympanic membrane and annulus are distinctive or partially concealed beneath thin skin on side of head. Dorsum finely shagreen to tuberculate, venter areolate, with rounded small tubercles; tubercles subconical or conical in the upper the eyelid and hells. Interdigital membranes absent and the Toe V is much longer than the Toe III. Tip of Toe V reaching distal border of distal subarticular tubercle of toe IV (Condition C). Fingers with lateral fringes or crenulations. Toes with lateral fringes or keels. Vocal slits and nuptial pads present or absent.

**Diversity** (Fig. 2). Fourteen species, seven formally described: *Pristimantis celator* (*Lynch, 1976*), *Pristimantis verecundus* (*Lynch & Burrowes, 1990*), *Pristimantis mutabilis* (*Guayasamin et al., 2015*), *Pristimantis broaddus* sp. nov., *Pristimantis praemortuus* sp. nov., *Pristimantis robayoi* sp. nov., *Pristimantis satheri* sp. nov., and seven candidate species (*P.* sp. 4. *P.* sp. 6.–*P.* sp.10., Figs. 1–2).

**Distribution** (Fig. 3). Restricted to the Andean montane forest slopes of northwestern Ecuador (Carchi, Imbabura, Pichincha, and Cotopaxi provinces) and southwestern Colombia (Nariño Department), between the Esmeraldas and Mira river basins, at elevation

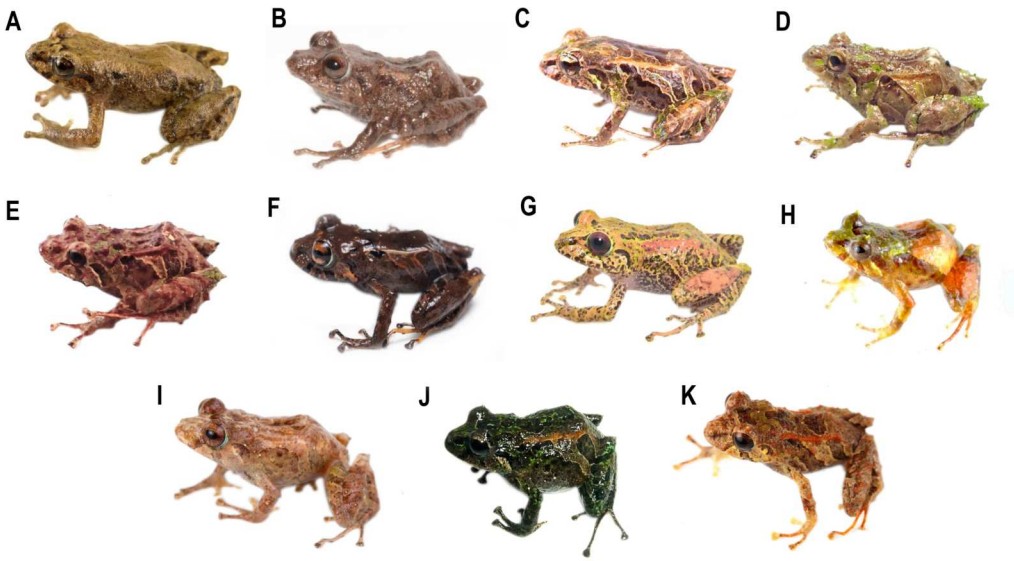

**Figure 2 Some species of *Pristimantis celator* clade.** (A) *P. celator* QCAZ66230, Maldonado, Carchi; (B) *P. verecundus* DHMECN 12599, Cerro Oscuro, Reserva Dracula, Carchi; (C) *P. mutabilis*, QCAZ 76813, Los Cedros, Imbabura; (D) *P. broaddus* sp. nov., DHMECN 19037, Holotype, Cabañas el Pailón, Bosque Protector Cerro Golondrinas; (E) *P. praemortuus* sp. nov., DHMECN 19557, Holotipo, Bloque 20, Reserva Dracula, Carchi; (F) *P. robayoi* sp. nov., DHMECN 17894, holotype, Cerro Negro, Reserva Dracula, Carchi; (G) *P. satheri* sp. nov., DHMECN 14858, holotype, Chinambí, Carchi; (H) *P.* c.sp.4, DHMECN 16570, El Pailón, Reserva Dracula. Carchi; (I) *P.* c.sp.9, no collected specimen, Reserva Manduriacu, Imbabura; (J) *P.* c.sp.10, no collected, Estación Experimental La Favorita, Pichincha; (K) *P.* c.sp.8, no collected specimen, Reserva Taira, Esmeraldas. Not scale. Photographs by Mario H. Yánez Muñoz (B, J), Christian Paucar V. (D, E), Julio C. Carrión (F), Santiago R. Ron (A, C), Mateo Vega-Yánez (G), Jaime Culebras (H, I, J).

ranges from 500 m to 2,200 m elevation. Inhabit in Western foothill forest, Western Low Montane Forest, and Western montane forest ecosystems (*MAE, 2013*).

**Comments.** Originally *Hedges, Duellman & Heinicke (2008)* determined the phylogenetic position of *Pristimantis verecundus* from material collected in Cotopaxi province (QCAZ 12410) assigning it to the *P. unistrigatus* species group. Based on this published sequence, *Padial, Grant & Frost (2014)* do not include it in any species group. *Franco-Mena et al. (2023)* identify the *Pristimantis verecundus* clade in polytomy with the *Pristimantis myersi* group, although, *Guayasamin et al. (2015)* previously reported the monophyly of these clades. It also determines that the sequence used by *Hedges, Duellman & Heinicke (2008)* and *Padial, Grant & Frost (2014)* corresponds to a new candidate species, assigns *sensu stricto* sequences of *Pristimantis verecundus,* and determines high cryptic diversity. In this work, we reorganize the *Pristimantis verecundus* clade redefined it as the *Pristimantis celator* clade, and assign this clade as part of the *Pristimantis myersi* species group. We consider the polytomy between the *Pristimantis jubatus*+*P. celator* clades with the *Pristimantis myersi* group are due to the small size of the available sequences of *Pristimantis jubatus*. We confirm the close relationship between the *Pristimantis myersi* species group and the *Pristimantis leptlophus* +*P.boulengeri*+ *P. devillei* groups, therefore,

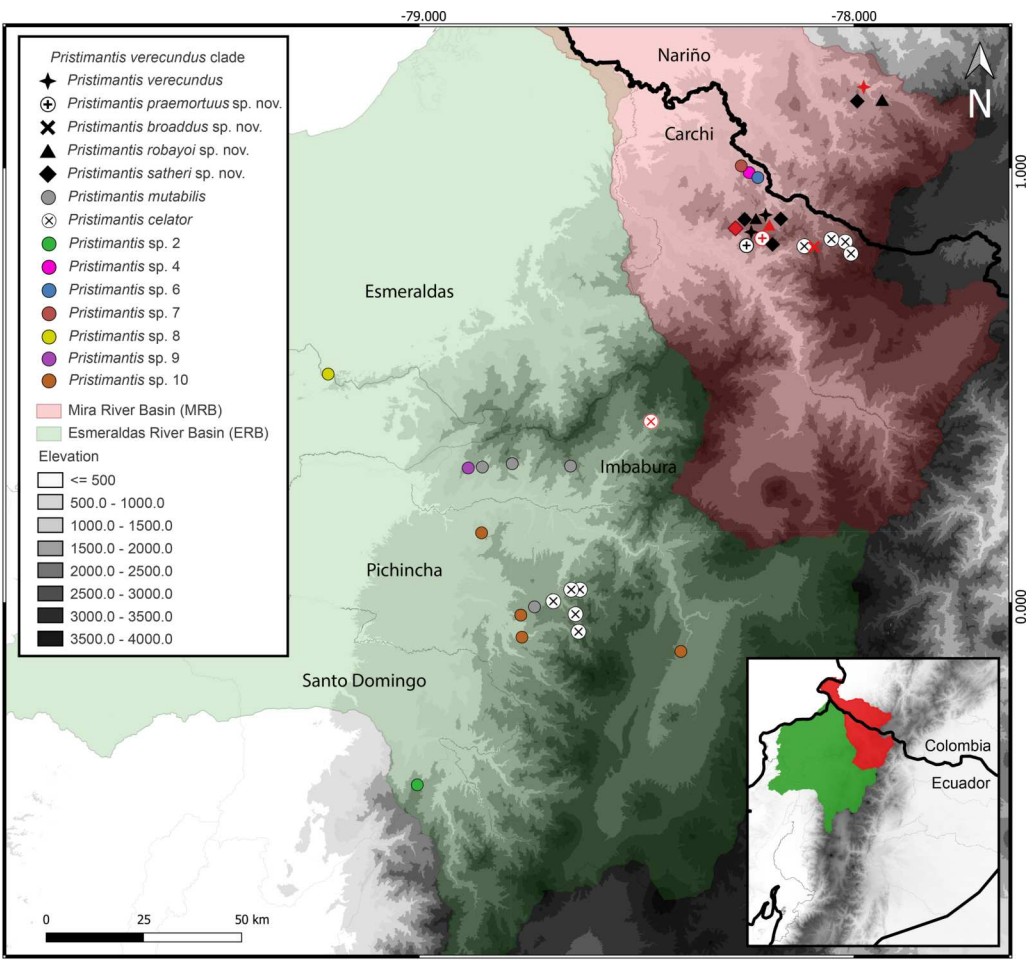

**Figure 3** **Map of distribution of *Pristimantis celator* clade.** Type localities are shown in red colors for different symbols. Dots with different colorations correspond to candidate species. Symbology for the formalin species described is indicated in the map legend. Map by: Julio C. Carrión-Olmedo.

like other authors, we suggest that this large, high-support clade should be considered the subgenus *Trachyphrynus* (*Hedges, Duellman & Heinicke, 2008*; *Rivera-Correa & Daza, 2016*; *González-Durán et al., 2017*; *Jetz & Pyron, 2018*; *Bejarano-Muñoz et al., 2022*).

## New species

*Pristimantis praemortuus* **sp. nov.**
*Pristimantis* sp.5 *Franco-Mena et al., 2023*.
**LSIDurn: lsid:zoobank.org:act:0C3E9CF7-3D61-4440-8203-C3E4A5D582C0**
*Common name in Spanish: Cutín previo a la muerte*
*Suggested common English name: Praemortuus's rainfrog*

**Holotype** (Figs. 2, 4–9). DHMECN 19591, adult female, from the Bloque 20, Sector Los Olivos (Bloque 20), Dracula Reserve, Tulcán, Carchi province, Ecuador, (0.84349, -78.21272; 2,339 m), collected on 13 December 2023 by Mario Humberto Yánez-Muñoz, Christian Paucar Veintimilla, Carlos Ríos & Miguel Urgilés-Merchán.

**Paratype** (Fig. S4). A total of seven specimens were collected in the same type locality and data collectors. Adult females (2): DHMECN 19557 and DHMECN 19570 (0.83239, -78.24377; 2,390 m) on 08 December 2023. Adult males (3): DHMECN 19535 (0.83829, -78.22830; 2,379 m); DHMECN 19546 and DHMECN 19547, (0.83572, -78.23136; 2,288 m) DHMECN 19577, (0.83989, -78.22341; 2,357 m); collected on 08 December 2023 and one juvenile DHMECN 19579, (0.84025, -78.22314; 2,351 m), collected on 12 December 2023.

**Diagnosis.** *Pristimantis praemortuus* sp. nov. is a member of the *Pristimantis myersi* species group and *P. celator* species clade, characterized by the following combination of characters: (1) dorsal skin shagreen with scattered low tubercles throughout the body; flanks with subconical tubercles aggregated in the axillary region; small subconical tubercles on the ventral edge of the lower jaw; dorsolateral fold present well-defined, extending from the scapular region to or slightly beyond the sacral region (illium), the dorsal surface of the tibia covered by transverse rows of glandular tissue, discoidal fold present slightly visible, venter areolate with tiny scattered tubercles; (2) tympanum present, visible, tympanic annulus clearly defined, round, posterior border of tympanic annulus with a thick fold, horizontal diameter of tympanum equal to 46.87% of eye diameter ($n = 8$), antero-dorsal margin with a supratympanic fold composed by low warts, subconical postrictal tubercles present; (3) snout short, broadly rounded in dorsal view, curved, tip of snout protruding in lateral view; (4) upper eyelid with one small conical tubercle and with several lower subconical tubercles; cranial crest absent; (5) dentigerous processes of vomers positioned posterior to level of choanae, obliques in posterior outline, minute and embedded in roof of mouth, barely visible and therefore sometimes considered to be absent; (6) males unknown; (7) finger I shorter than finger II; discs broad and expanded mainly in the fingers II–IV, with circunmarginal grooves; (8) fingers with lateral fringes, weakly defined; (9) ulnar tubercles present, subconical placed on the outer edge of the ulna; (10) heel with an elongated conical tubercle surrounded by lower rounded tubercles, outer edge of the tarsus with a row of subconical tubercles, internal tarsal fold weakly defined; (11) metatarsal tubercle 2X larger than the external one which is rounded and subconical; few supernumerary tubercles and barely visible; (12) toes with weakly defined lateral fringes, basally pronounced, interdigital membrane absent, outer edge of the foot and toe IV crenulated, toe V longer than toe III, does not reach the base of the distal subarticular tubercle of the toe IV; (13) dorsal color pattern grayish olive to jet black (300) (chestnut to maroon in life), ventrally marmoleated with jet black to medium neutral gray with an inverted trapezoidal mark (throat with a distinctive incomplete trapezoidal burn umber in life). Hidden surfaces of thighs, legs, and groin are pinkish in preservative (reddish in life). Iris coppery gold to copper; (14) small adult males, SVL = 12.46–16.09 mm (mean = 14.28 mm, $n = 5$), females SVL = 17.03–17.68 mm (mean = 17.35 mm, $n = 3$).

**Comparison with other species** (Figs. 2, 5, 6, 7, 8). The new species and its closest sister species (*Pristimantis broaddus* sp. nov.) are the only species on the *P. celator* clade in having

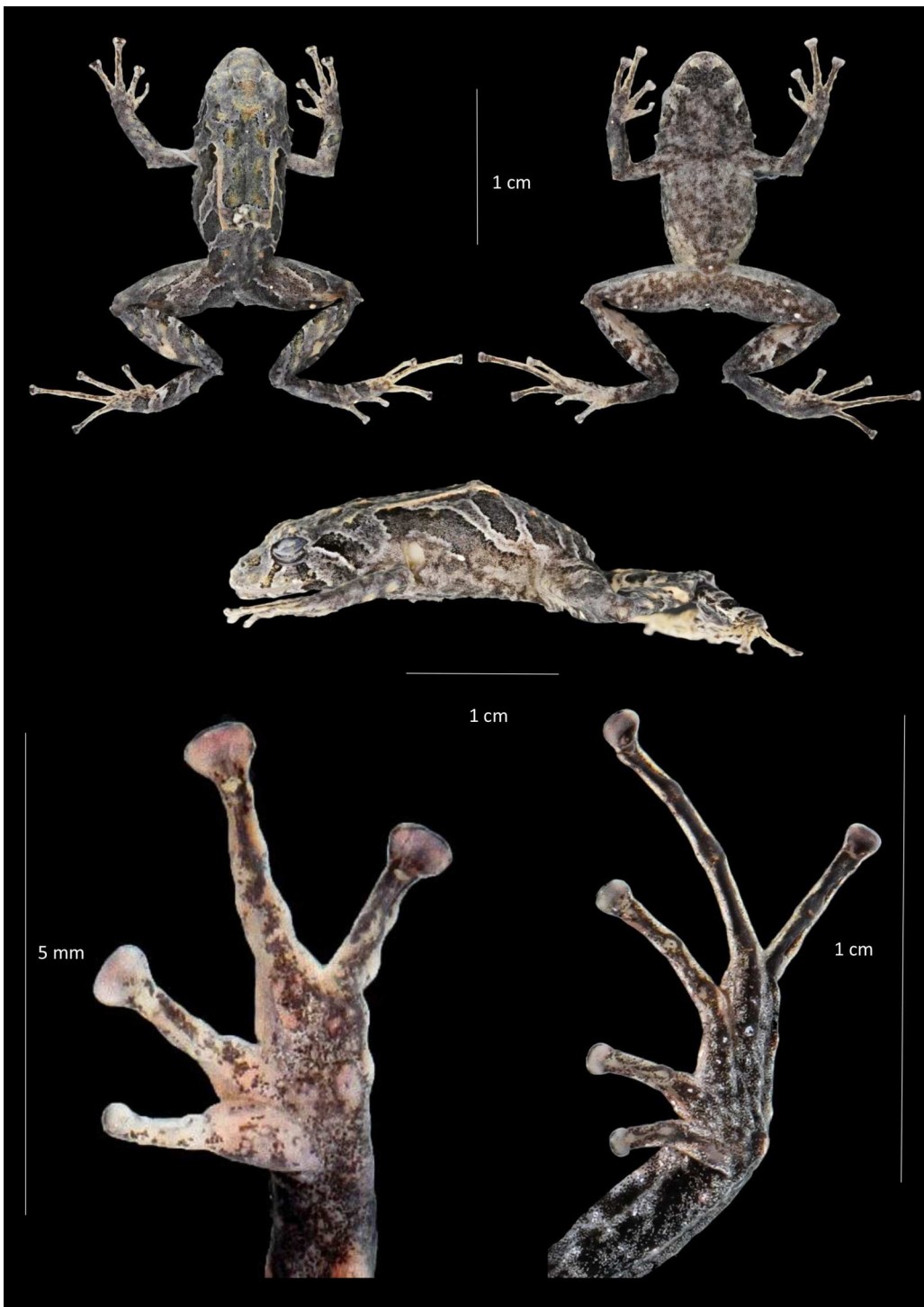

**Figure 4** **Dorsal, ventral and profile views of *Pristimantis praemortuus* sp.nov., (Holotype DHMECN 19591, female adult) and detail of hand and foot.** Photographs Christian Paucar V.

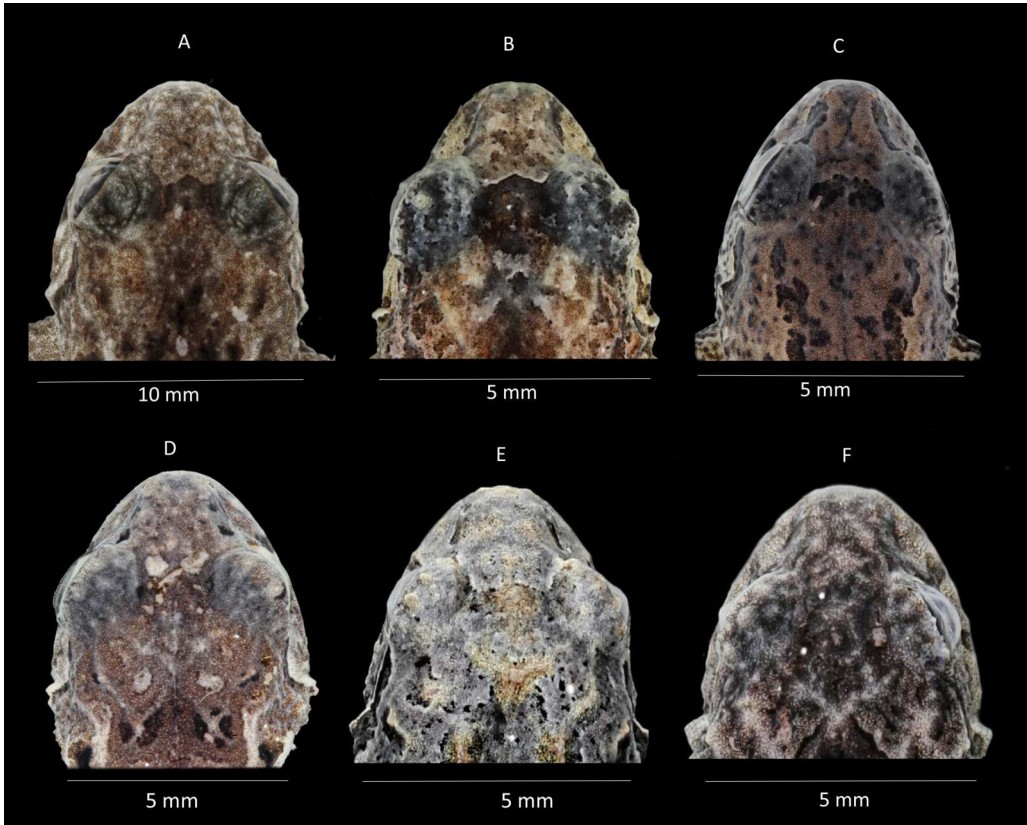

**Figure 5** **Comparison of heads in dorsal view of preserved of some species of *Pristimantis celator* clade.** (A) *P. verecundus*, Holotype IAvH 1801; (B) *Pristimantis mutabilis* DHMECN 11755; (C) *Pristimantis satheri* sp. nov., Holotype DHMECN 14858; (D) *Pristimantis robayoi* sp. nov., Paratype DHMECN 17894; (E) *Pristimantis praemortuus* sp. nov., Holotype DHMECN 19591; (F) *Pristimantis broaddus* sp.nov., Holotype DHMECN 19037. Photographs Christian Paucar V.

narrow digits of toes. *Pristimantis praemortuus* sp. nov. differs from the closely related species (*P. verecundus*, *P. mutabilis*, *P. robayoi* sp. nov., *P. satheri* sp. nov., and *P. broaddus* sp. nov., Fig. 1) by the presence of dorsolateral fold well-defined visible from the scapular region to or slightly beyond the sacral region (ilium) (complete dorsolateral glandular folds extending to the first quarter of the ilium in *P. satheri* sp. nov., dorsolateral fold reaching just before the sacrum in *P. verecundus*, dorsolateral fold only reach the level of sacrum in *P. mutabilis*, dorsolateral folds extending the entire length of the ilium in *P. robayoi* sp. nov.); throat with an inverted trapezoidal mark (v-shaped mark in *P. verecundus*, reduced or inconspicuous in *P. mutabilis* and absent in *P. robayoi* sp. nov. and *P. satheri* sp. nov.); the dorsal surface of the tibia covered by transverse rows of glandular tissue in *P. broaddus* sp. nov., *P. verecundus* and *P. praemortuus* sp. nov. (outlining the extreme edge of the heel in *P. robayoi* sp. nov., dorsal surface of tibia covered by glandular tissue in *P. satheri* sp. nov); and iris gold red in *P. broaddus* sp. nov. (bright orange-copper with a black edge in *P. robayoi* sp. nov., coppery gold in *P. praemortuus*, sp. nov. iris cream to golden with thin black reticulation and a reddish-brown horizontal streak in *P. mutabilis*, and wine-colored

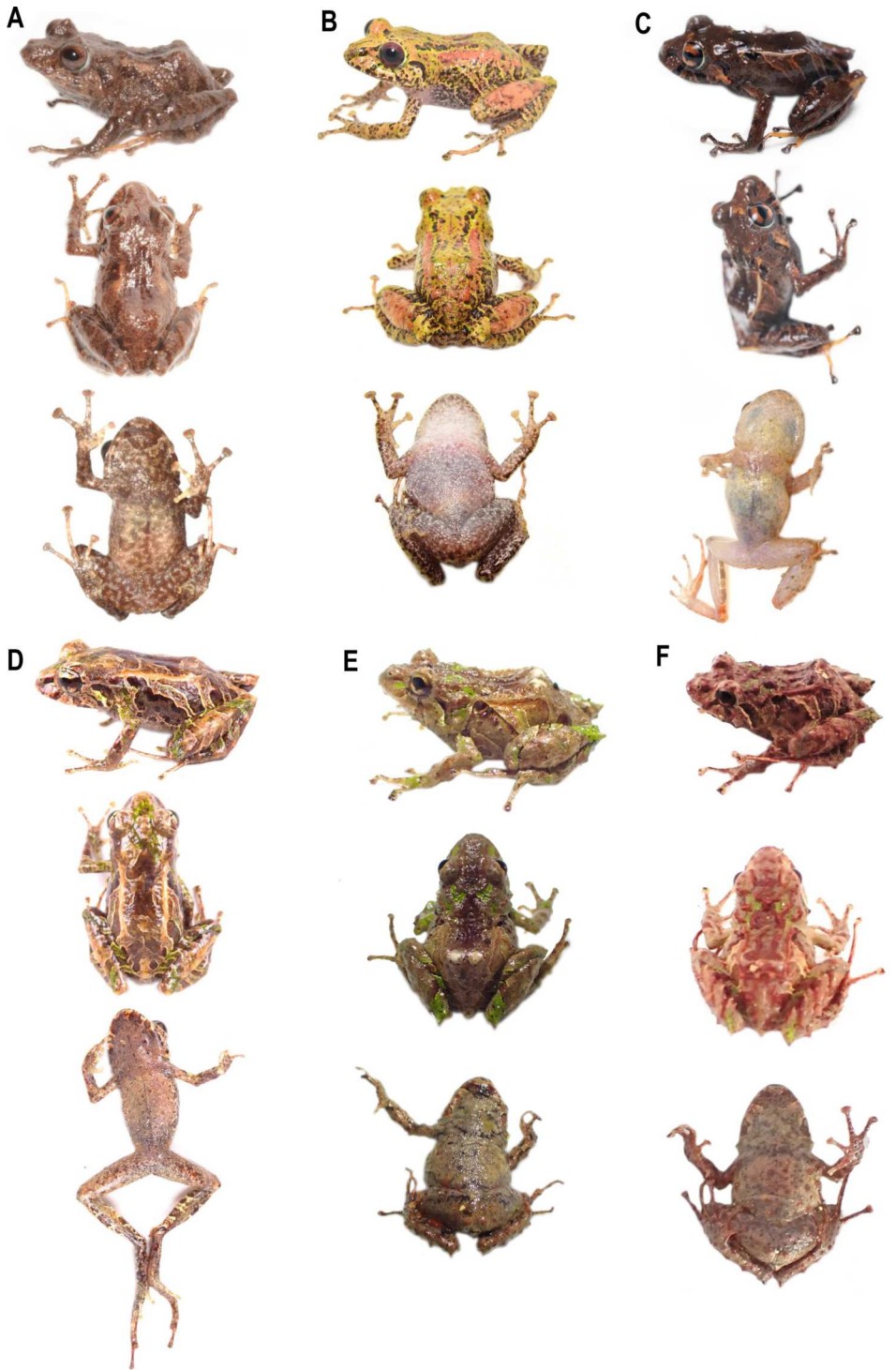

**Figure 6 Comparison of similar species of *Pristimantis celator* clade in life.** (A) *P. verecundus* DHMECN 12599, male, SVL: 18.17 mm; (B) *P. satheri* sp. nov. DHMECN 14858 Holotype, female, SVL: 23.74 mm; (C) *P. robayoi* sp. nov., DHMECN 17984 Holotype, male, SVL: 24.22 mm; (D) *P. mutabilis*, QCAZ 76813; (E) *P. broaddus* sp. nov., DHMECN 19029, female, SVL: 18.21 mm; (F) *P. praemortuus* sp.nov., DHMECN 19591, female, SVL: 17.58 mm. Photographs by Mario H. Yánez Muñoz (A), Christian Paucar V. (E, F), Julio C. Carrión (C), Santiago R. Ron (D).

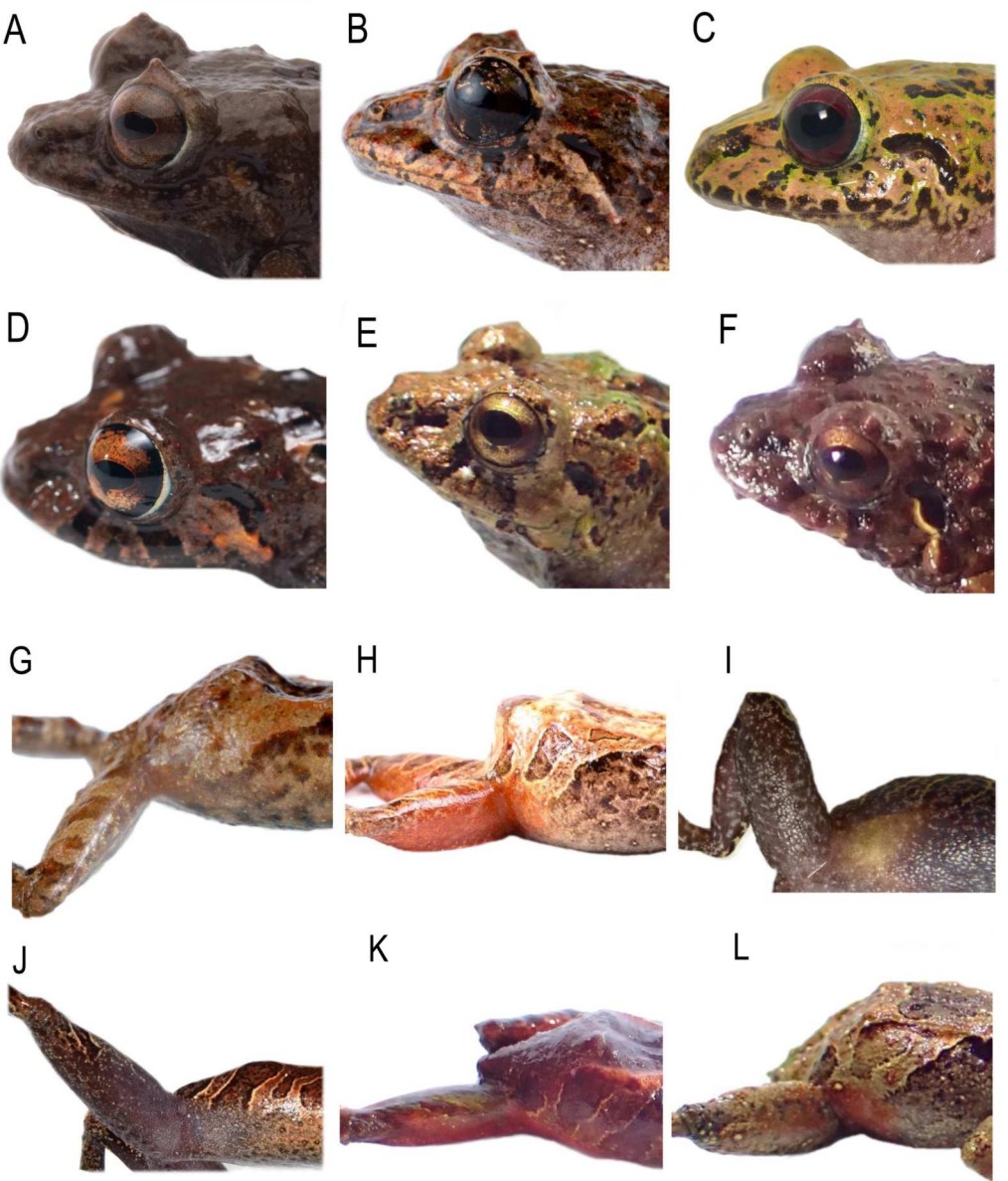

**Figure 7** **Comparison of heads profiles views (A–B) and groin coloration (G–L) in life of similar species of *Pristimantis celator* clade.** (A, G) *P. verecundus* DHMECN 17892; (B, H) *P. mutabilis*, no collected specimen; (C, I) *P. satheri* sp. nov. DHMECN 14858 Holotype; (D) *P. robayoi* sp. nov., DHMECN 17984 Holotype; (E, K) *P. broaddus sp.nov.* DHMECN 19029; (F, L) *P. praemortuus* sp.nov. DHMECN 19591. Photographs by Mario H. Yánez Muñoz (A), Christian Paucar V. (E, F, K, L), Julio C. Carrión (A, D), Santiago R. Ron (H), Jaime Culebras (G); Mateo Vega-Yánez (C, I).

in *P. satheri* sp. nov.). Tympanic annulus clearly defined, round, posterior border of tympanic annulus with a thick fold in *P. praemortuus* sp. nov. (tympanic annulus weekly defined in *P. broaddus* sp. nov.), fingers with lateral fringes weakly defined in *P. praemortuus* sp. nov. (strongly crenulated mainly in finger IV lateral fringes in *P. broaddus* sp. nov.),

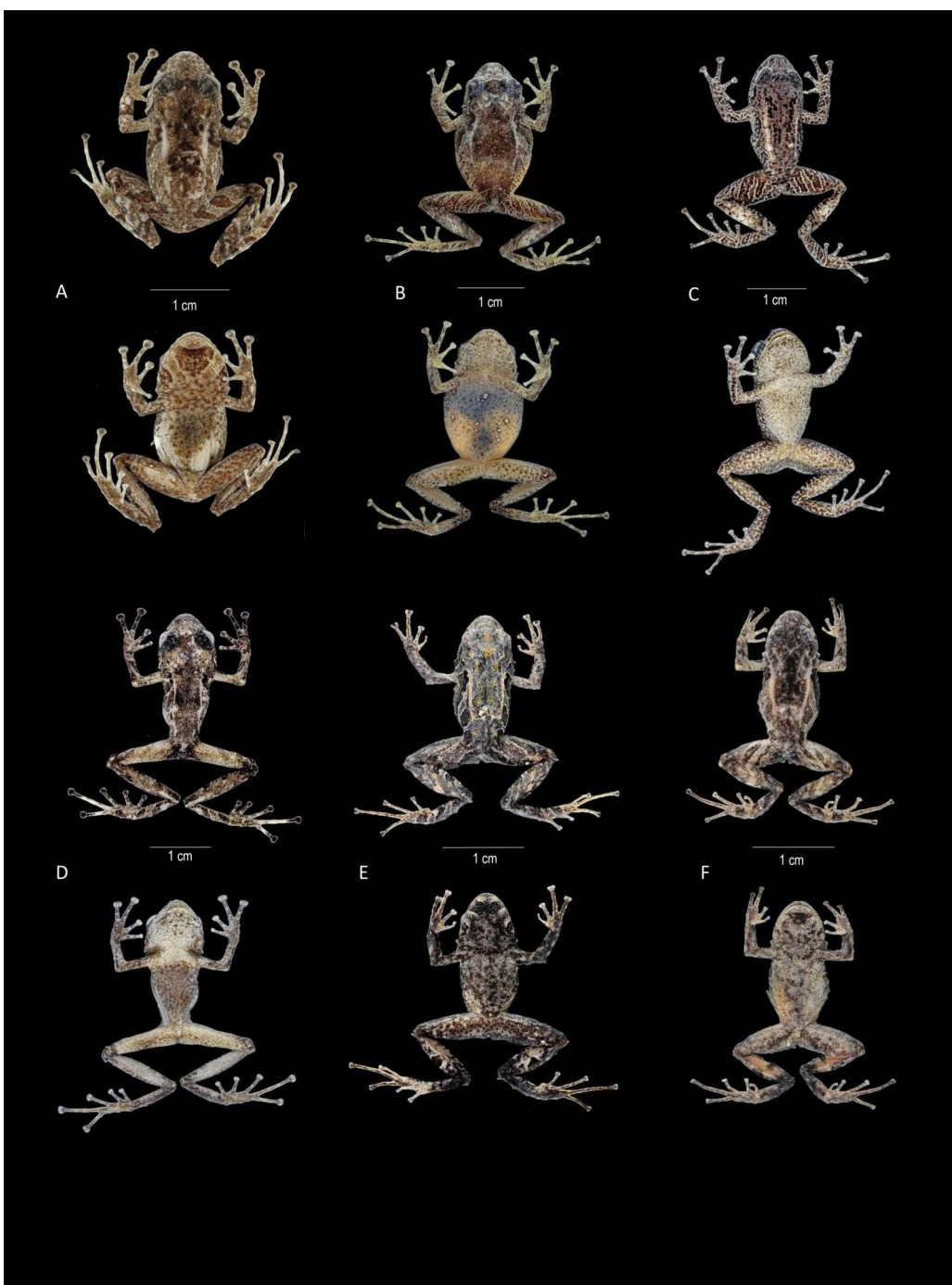

**Figure 8 Comparison in preservative of similar species of *Pristimantis celator* clade.** (A) *Pristimantis verecundus*, Holotype IAvH 1801; (B) *Pristimantis mutabilis* DHMENC 11755; (C, G) *Pristimantis satheri* sp. nov. Holotype DHMECN 14858; (D, E) *Pristimantis robayoi* sp. nov., Holoype DHMECN 17894; (E) *Pristimantis praemortuus* sp. nov., Holotype DHMECN 19591; (F) *Prismtimantis boaddus* sp. nov., Holotype DHMENC 19037. Photographs Christian Paucar.

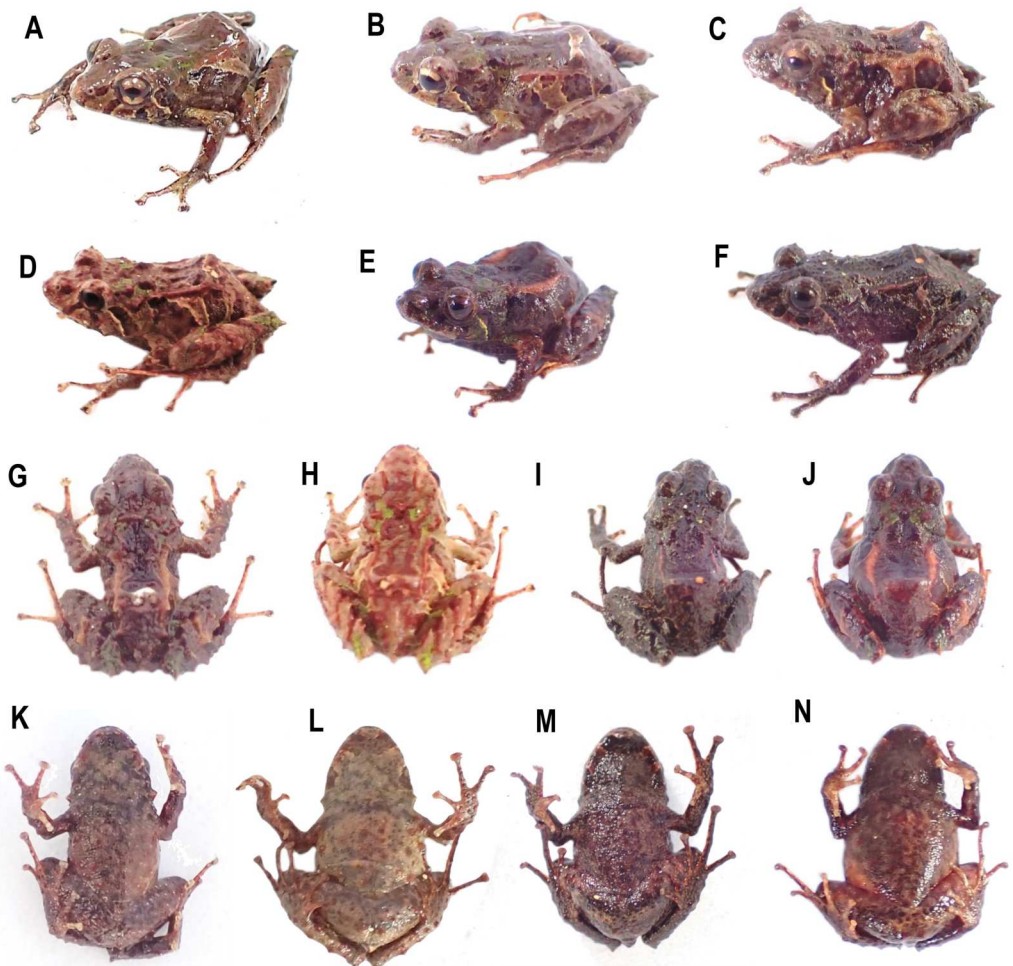

**Figure 9** Variation in life of the type series of *Pristimantis praemortuus* sp. nov. (A) DHMECN 19535, male, SVL: 16.39 mm; (B) DHMECN 19547, male, SVL: 14.76 mm; (C, G, K) DHMECN 19546, male, SVL: 12.46 mm; (D, H, L) DHMECN 19591 Holotype, female, SVL: 17.68 mm; (E, J, N). DHMECN 19557, female, SVL: 17.33 mm; (F, I, M) DHMECN 19570, female, SVL: 17.03 mm. Photographs by Christian Paucar.

toes with weakly defined lateral fringes, basally pronounced, interdigital membrane absent, outer edge of the foot and toe IV crenulated in *P. praemortuus* sp. nov. (toes with weakly defined lateral fringes, basally pronounced, interdigital membrane absent in *P. broaddus* sp. nov.).

Genetic distances with close-related species (*Pristimantis praemortuus* sp. nov. *vs. P. broaddus* sp. nov.) have 3.54%. The genetic distances with other species in this clade range from 3.54%–12.49% (Table S2).

**Description of the holotype** (Fig. 4). Adult female (DHMECN 19591), head slightly wider than long. Snout short, broadly rounded in dorsal view, short, curved, tip of snout protruding in lateral view with low rounded scattered tubercles; eye-nostril distance 9.44% of SVL, canthus rostralis slighthly rounded in cross section, in dorsal view weakly concave,

loreal region concave, slightly protruding narines directed dorsolaterally; interorbital area with a low rounded interorbital tubercule, interorbital fold absent, interorbital distance wider than upper eyelid, 80.45%; cranial crest absent, low dermal fold in the occipital and prootic region; upper eyelid with one small conical tubercle and with several lower subconical tubercles (reduced by preservation effects); cranial crest absent, tympanum present, visible, tympanic annulus clearly defined, round, posterior border of tympanic annulus with a thick fold, horizontal diameter of tympanum equal to 49.5% of eye diameter, antero-dorsal margin with a supratympanic fold composed by low warts, subconical postrictal tubercles present, diameter of tympanum equals 45.83% of eye diameter, underside of tympanum with large two or three subconical postrictal tubercles, subconical; choanae small, oval in outline, not covered by palatal floor of maxilla; dentigerous processes of vomers positioned posterior to level of choanae, oblique in posterior outline, minute and embedded inf roof of mouth, barely visible, no visible teeth, tongue as wide as long, oval, 40% attached to floor of mouth.

Texture of dorsum shagreen with scattered low tubercles throughout the body; flanks with subconical tubercles aggregated in the axillary region; small subconical tubercles on the ventral edge of the lower jaw; dorsolateral fold present well-defined, extending from the scapular region to or slightly beyond the sacral region (ilium), discoidal fold present slightly visible, venter areolate with tiny scattered tubercles; two evident ulnar tubercles present, subconical placed on the outer edge of the ulna; outer palmar tubercle lobular, segmented into three parts, approximately two times the size of elongated outer thenar tubercle; supernumerary tubercles present; subarticular tubercles defined, rounded in lateral view. Fingers with lateral fringes, weakly defined, discs about $1.8\times$ the size of the digit. Hind limbs slender (TL 45.47% of SVL; FL 48.99% of SVL); heel with an elongated conical tubercle surrounded by lower rounded tubercles, outer edge of the tarsus with a row of subconical tubercles, internal tarsal fold weakly defined; metatarsal tubercle larger than the external one which is rounded and subconical; few barely visible supernumerary tubercles; toes have weakly defined lateral fringes that are more pronounced at the base, lack an interdigital membrane, and exhibit crenulation along the outer edge of the foot and toe IV. Toe V is longer than toe III but does not extend to the base of the distal subarticular tubercle of toe IV.

**Holotype coloration in preservative** (Fig. 4). Dorsum grayish olive (274) with scattered irregular yellowish-green marks and distinctive grayish olive (274) stripes on the flanks bordered in cream, with orange-cream dorsolateral folds. Limbs have grayish olive (274) transverse bands separated by creamy white interspaces. Venter creamy gray marbled with grayish brown (284), with a distinctive grayish brown (284) trapezoidal mark and other irregular grayish brown (284) marks on the throat.

**Holotype coloration in life** (Fig. 9). Dorsal coloration chestnut (30), with distinctive cinnamon (21) diagonal stripes on the flanks, bordered with cream. Dorsolateral folds burnt umber (48). Chestnut (30) supratympanic stripe, solid and complete, reaching the postrictal tubercle, with cream anterior border. Labial stripes and transversal tibial (glandular) folds chestnut (30). Outer heel edge leaf green (122). Iris copper. Ventrally
**Table 2  Morphometric characters of the news species and *Pristimantis vercundus*..**

| Character | *P. satheri* sp. nov. Females n = 2 | Males n = 2 | *P. broaddus* sp. nov. Females n = 8 | *P. praemortuus* sp. nov. Females n = 3 | Males n = 5 | *P. robayoi* sp. nov. Females n = 7 | Males n = 3 | *P. verecundus* Females n = 4 | Males n = 3 |
|---|---|---|---|---|---|---|---|---|---|
| SVL | 22.08–23.74 (2) 22.91 ± 1.17 | 19.73–19.79 (2) 19.76 ± 0.04 | 16.81–18.42 (8) 17.62 ± 1.14 | 17.03–17.68 (3) 17.35 ± 0.46 | 12.46–17.4 (5) 14.93 ± 3.49 | 21.78–29.05 (7) 25.42 ± 5.14 | 18.08–23.32 (3) 20.7 ± 3.71 | 18.57–22.84 (4) 20.71 ± 3.02 | 17.56–19.24 (3) 18.4 ± 1.19 |
| HW | 7.74–8.51 (2) 8.13 ± 0.54 | 6.95–6.99 (2) 6.97 ± 0.03 | 5.28–6.22 (8) 5.75 ± 0.66 | 5.99–6.16 (3) 6.16 ± 0.09 | 5.13–5.99 (5) 5.56 ± 0.61 | 7.68–9.81 (7) 8.75 ± 1.51 | 6.31–8.61 (3) 7.46 ± 1.63 | 7.15–8.76 (4) 7.96 ± 1.14 | 6.26–7.30 (3) 6.78 ± 0.74 |
| HL | 8.76–9.23 (2) 9 ± 0.33 | 7.6–7.62 (2) 7.61 ± 0.01 | 5.62–6.26 (8) 5.94 ± 0.45 | 5.26–6.2 (3) 5.66 ± 0.57 | 4.43–5.6 (5) 5.02 ± 0.83 | 6.7–9.51 (7) 8.11 ± 1.99 | 5.3–7.42 (3) 6.36 ± 1.5 | 5.9–7.35 (4) 6.63 ± 1.03 | 5.12–6.52 (3) 5.82 ± 0.99 |
| EN | 2.15–2.65 (2) 2.4 ± 0.35 | 1.99–2.1 (2) 2.05 ± 0.08 | 1.58–2.75 (8) 2.17 ± 0.83 | 1.71–1.86 (3) 1.73 ± 0.08 | 1.31–1.97 (5) 1.64 ± 0.47 | 2.19–3.17 (7) 2.68 ± 0.69 | 1.9–2.61 (3) 2.26 ± 0.5 | 1.72–2.4 (4) 2.06 ± 0.48 | 1.53–2.04 (3) 1.79 ± 0.36 |
| IND | 1.77–1.91 (2) 1.84 ± 0.1 | 1.56–1.6 (2) 1.58 ± 0.03 | 1.64–1.91 (8) 1.78 ± 0.19 | 1.74–1.75 (3) 1.76 ± 0.03 | 1.34–1.7 (5) 1.52 ± 0.25 | 1.73–2.28 (7) 2.01 ± 0.39 | 1.69–1.84 (3) 1.77 ± 0.11 | 1.69–2.1 (4) 1.9 ± 0.29 | 1.63–1.87 (3) 1.75 ± 0.17 |
| IOD | 2.6–2.62 (2) 2.61 ± 0.01 | 2.12–2.21 (2) 2.17 ± 0.06 | 1.69–2.43 (8) 2.06 ± 0.52 | 1.79–2.15 (3) 1.86 ± 0.09 | 1.6–1.95 (5) 1.78 ± 0.25 | 2.23–2.96 (7) 2.6 ± 0.52 | 1.84–2.5 (3) 2.17 ± 0.47 | 2–2.42 (4) 2.21 ± 0.3 | 1.93–2.19 (3) 2.06 ± 0.18 |
| EW | 1.56–2 (2) 1.78 ± 0.31 | 1.39–1.4 (2) 1.4 ± 0.01 | 1.2–1.52 (8) 1.36 ± 0.23 | 1.36–1.47 (3) 1.43 ± 0.09 | 1.12–1.75 (5) 1.44 ± 0.45 | 1.79–3.04 (7) 2.42 ± 0.88 | 1.67–2.15 (3) 1.91 ± 0.34 | 1.38–2.32 (4) 1.85 ± 0.66 | 1.42–1.77 (3) 1.6 ± 0.25 |
| TD | 0.97–1.06 (2) 1.02 ± 0.06 | 0.9–0.95 (2) 0.93 ± 0.04 | 0.88–1.11 (8) 1 ± 0.16 | 0.89–0.95 (3) 1.02 ± 0.12 | 0.8–1.09 (5) 0.95 ± 0.21 | 0.7–1.12 (7) 0.91 ± 0.3 | 0.76–1.22 (3) 0.99 ± 0.33 | 0.69–0.91 (4) 0.8 ± 0.16 | 0.77–0.98 (3) 0.88 ± 0.15 |
| ED | 2.83–3.14 (2) 2.99 ± 0.22 | 2.5–2.53 (2) 2.52 ± 0.02 | 1.97–2.48 (8) 2.23 ± 0.36 | 1.72–2.2 (3) 2.06 ± 0.48 | 1.79–2.15 (5) 1.97 ± 0.25 | 3.09–3.5 (7) 3.3 ± 0.29 | 2.83–3.26 (3) 3.05 ± 0.3 | 2.43–2.77 (4) 2.6 ± 0.24 | 1.87–2.66 (3) 2.27 ± 0.56 |
| TL | 11.05–11.63 (2) 11.34 ± 0.41 | 9.25–9.8 (2) 9.53 ± 0.39 | 7.79–8.44 (8) 8.12 ± 0.46 | 1.5–8.18 (3) 5.09 ± 5.07 | 6.54–8.95 (5) 7.75 ± 1.7 | 10.55–13.76 (7) 12.16 ± 2.27 | 9.72–13.31 (3) 11.52 ± 2.54 | 8.88–10.94 (4) 9.91 ± 1.46 | 8.45–9.6 (3) 9.03 ± 0.81 |
| HaL | 6.78–7.13 (2) 6.96 ± 0.25 | 5.6–5.92 (2) 5.76 ± 0.23 | 4.75–5.49 (8) 5.12 ± 0.52 | 4.73–4.97 (3) 5.05 ± 0.45 | 4.07–4.65 (5) 4.36 ± 0.41 | 6.52–8.3 (7) 7.41 ± 1.26 | 5.63–7.26 (3) 6.45 ± 1.15 | 5.23–6.57 (4) 5.9 ± 0.95 | 5.5–6.43 (3) 5.97 ± 0.66 |
| FoL | 11.05–11.86 (2) 11.46 ± 0.57 | 9.5–9.7 (2) 9.6 ± 0.14 | 2.12–9.26 (8) 5.69 ± 5.05 | 7.99–8.59 (3) 8.51 ± 0.73 | 6.58–7.71 (5) 7.15 ± 0.8 | 10.24–13.47 (7) 11.86 ± 2.28 | 8.97–10.64 (3) 9.81 ± 1.18 | 8.55–10.59 (4) 9.57 ± 1.44 | 8.79–9.9 (3) 9.35 ± 0.78 |
| F3D | 1.08–1.27 (2) 1.18 ± 0.13 | 0.88–0.89 (2) 0.89 ± 0.01 | 0.54–1.02 (8) 0.78 ± 0.34 | 0.64–0.85 (3) 0.76 ± 0.17 | 0.59–0.74 (5) 0.67 ± 0.11 | 1–1.44 (7) 1.22 ± 0.31 | 0.74–1.15 (3) 0.95 ± 0.29 | 0.99–1.38 (4) 1.19 ± 0.28 | 0.99–1.16 (3) 1.08 ± 0.12 |
| T4D | 0.98–1.04 (2) 1.01 ± 0.04 | 0.6–0.64 (2) 0.62 ± 0.03 | 0.41–0.81 (8) 0.61 ± 0.28 | 0.51–0.59 (3) 0.58 ± 0.04 | 0.47–0.76 (5) 0.62 ± 0.21 | 0.95–1.35 (7) 1.15 ± 0.28 | 0.59–1.24 (3) 0.92 ± 0.46 | 0.85–1.23 (4) 1.04 ± 0.27 | 0.68–0.76 (3) 0.72 ± 0.06 |

**Notes.**

Measurements: Snout–vent length (SVL), Head width (HW), Head length (HL), Horizontal eye diameter (ED), Interorbital distance (IOD) Eye–nostril distance (EN), Tympanic length (TD), Internarinal distance (IND), Tibia length (TL), Upper eyelid width (EW), Foot length (FoL), Hand length (HaL), Disc width of finger III (F3D), Disc width of toe IV (T4D).

robin rufous (29), mottled with burnt umber (48), with scattered burnt umber (48) tubercles, throat with a distinctive trapezoidal burnt umber (48) mark.

**Measurements (in mm) of holotype.** SVL = 17.68; HW = 6.22; HL = 5.37; ED = 2.4; IOD = 1.83; EN = 1.67; TD = 1.1; IND = 1.79; TL = 8.67; EW = 1.49; FoL = 9.02; HaL = 5.36; FW = 0.88; TW = 0.61.

**Variation**. *Pristimantis praemortuus* sp. nov. shows sexual dimorphism in body size (Fig. S3) and coloration (Fig. 6; Fig. S4). Males of this species between 0.6–0.8 times smaller in SVL than females (Fig. S3). Other morphometric variation in the type series is showing in the Table 2.

In life (Fig. 6), dorsal coloration chestnut (30) (DHMECN 19535, DHMECN 19547, DHMECN 19546, DHMECN 19591) to maroon (39) (DHMECN 19557, DHMECN 19570), with distinctive diagonal stripes on the flanks ranging from cinnamon (21) (DHMECN 19535, DHMECN 19547, DHMECN 19546, DHMECN 19591) to burnt sienna (DHMECN

19557, DHMECN 19570), bordered in cream. The dorsum may display speckled marks of leaf green (122) (DHMECN 19535, DHMECN 19570), dorsolateral folds reddish-brown (DHMECN 19557, DHMECN 19557) to carmine (64) (DHMECN 19546, DHMECN 19591), burnt umber (48) (DHMECN 19570), or ferruginous (35) (DHMECN 19557). Chestnut (30) supratympanic stripe, solid and complete, reaching the postrictal tubercle, with cream anterior border. Labial stripes and transversal tibial (glandular) folds chestnut (30). Outer heel edge leaf green (DHMECN 19546, DHMECN 19591, DHMECN 19570, DHMECN 19557). Iris coppery gold to copper. Ventrally robin rufous (29) (DHMECN 19546, DHMECN 19557) to burnt sienna (38) (DHMECN 19591, DHMECN 19570), speckled or mottled with burnt umber (48), with scattered burnt umber (48) or reddish-brown tubercles (DHMECN 19546), throat with a distinctive incomplete trapezoidal burn umber (48).

In preservative (Fig. S4), dorsal color pattern varies from grayish olive (274) (DHMECN 19591, 19547, 19538), medium neutral grey (299) (DHMECN 19547) to with jet black (300) (DHMECN 19535). Limbs coloration with oblique bars sepia (286) (DHMECN 19547) or army brown (46) (DHMECN 19546). Belly marmoleated with jet black (300), to medium neutral gray (298) (DHMECN 19591, 19546–47) to pale neutral gray (296) (DHMECN 19535, 19570) background. Colorations salmon (251) (DHMECN 19577) to pale neutral gray (12) (DHMECN 19535, 19570 19546) on the groins and thighs and metatarsal regions, fingers and toes.

**Etymology**. From the Latin "*praemortuus*", meaning: "before dying". The specific epithet is an adjective that emphasizes the urgency of taxonomic efforts in conservation, highlighting the importance of describing species before they disappear or become extinct.

**Distribution and natural history.** *Pristimantis praemortuus* sp. nov. is known from the type locality in the Chinambí sector and on the western slope of Cerro Golondrinas of the Dracula Reserve (Block 20) in the province of Carchi, between 2,176 to 2,390 m of elevation, in the river basin (Fig. 3). This species is found in the montane evergreen forest of the Western Cordillera of the Andes (*MAE, 2013*), characterized by a closed canopy with trees up to 20 m high, covered by epiphytes, orchids, bromeliads, bryophytes, and ferns. The nine known specimens of *Pristimantis praemortuus* sp. nov. were found active at night perching or vocalizing on fern leaves and bushes in the lower and middle stratum of the forest, between 20 and 210 cm high. They were found in sympatry with: *P. pteridophilus* complex, *P. apiculatus*, *P. hectus*, *P.* grp. *devillei* and *P. satheri* sp. nov.

*Pristimantis broaddus* sp. nov.
LSIDurn:lsid:zoobank.org:act:F060D100-C4EA-4882-BCFE-5EEBD8D9FFB0
*Common name in Spanish*: *Cutín de Callie Broaddus*
*Suggested common English name*: *Broaddus'rainfrog*

**Holotype** (Figs. 2, 5, 6, 7, 8, 10, 11). DHMECN 19037, adult female, from the Bosque Protector Golondrinas, El Pailón, El Goltal, Espejo, Carchi province, Ecuador, (0.81833,

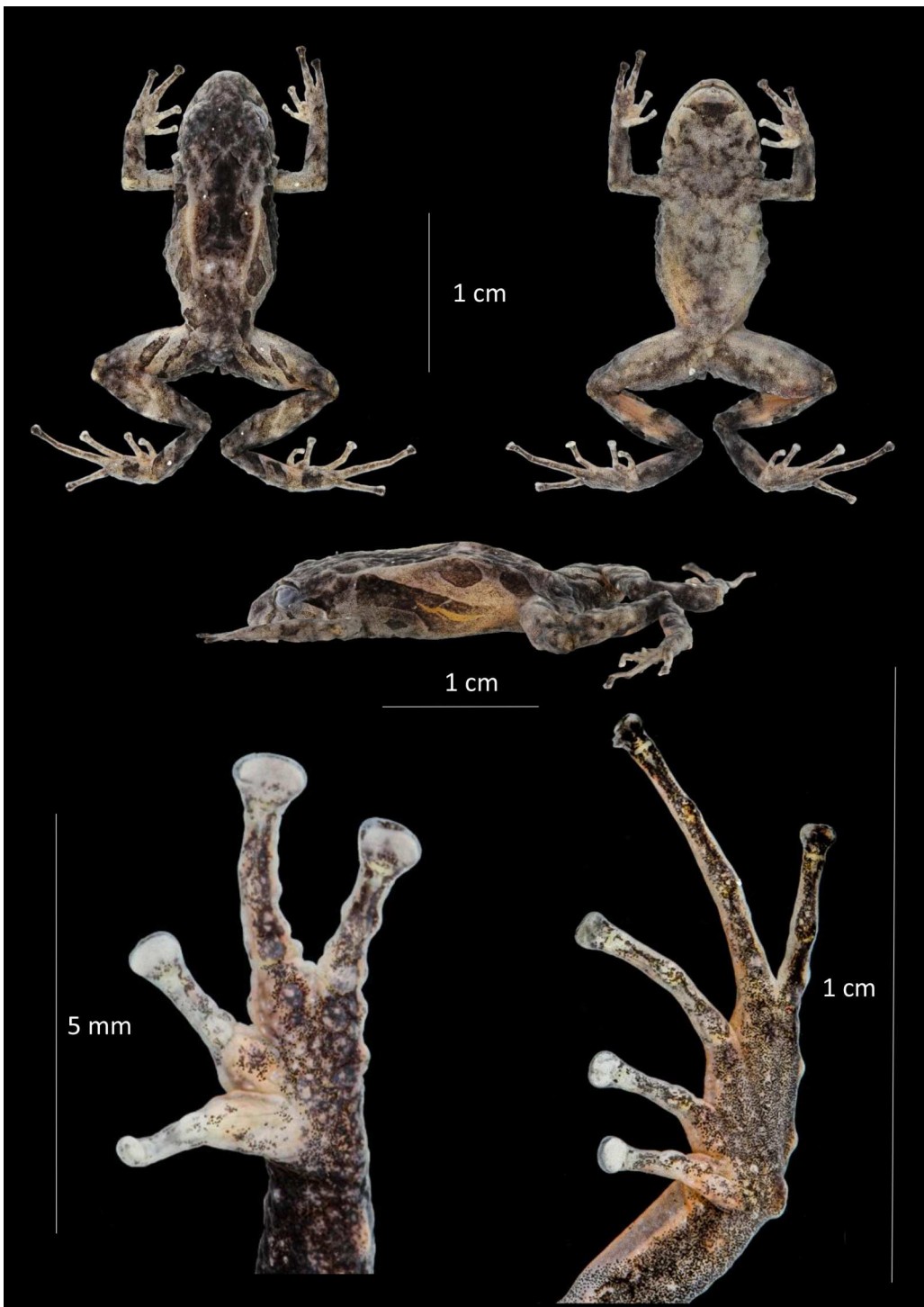

**Figure 10 Dorsal, ventral and profile views of *Pristimantis broaddus* sp. nov., female adult (DHMECN 19037) and detail of hand and foot.** Photographs Christian Paucar V.

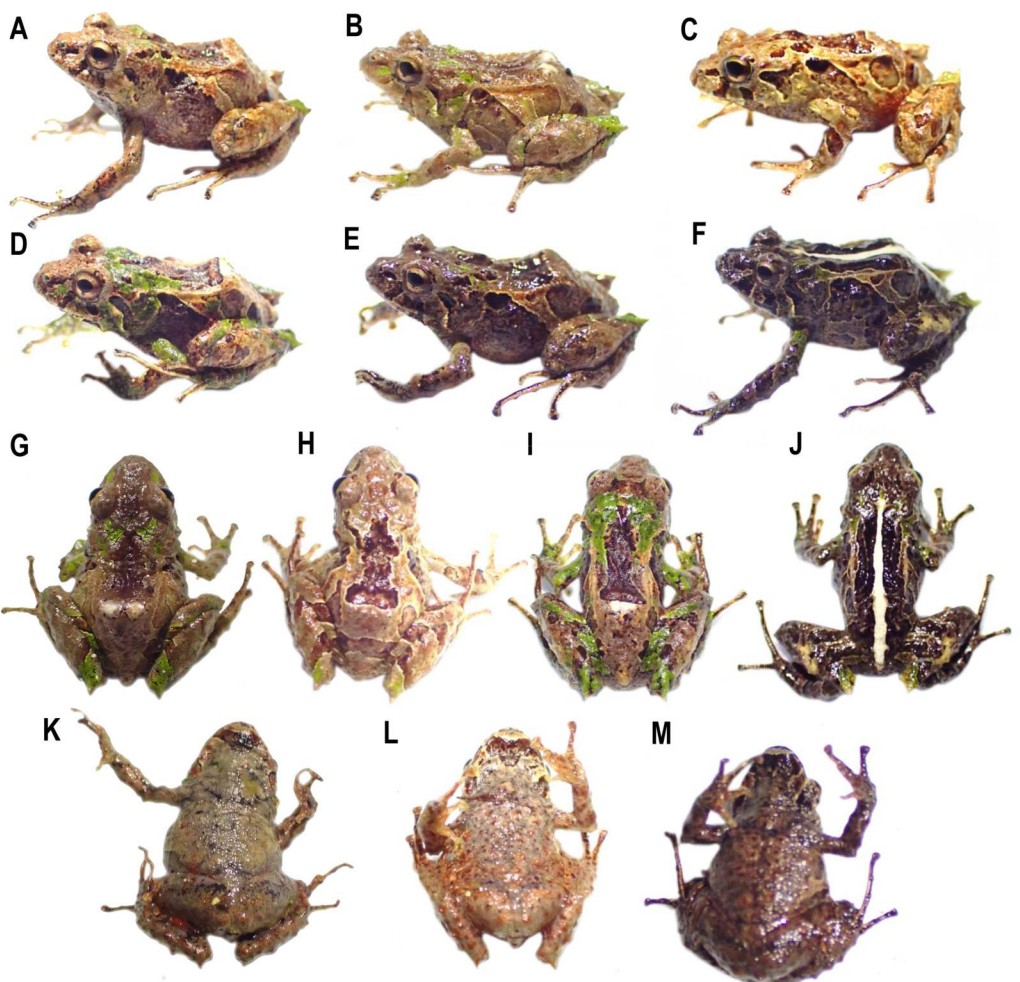

**Figure 11** **Variation in life of the type series of *Pristimantis broaddus* sp. nov.** (A) DHMECN 19029, female, SVL: 18.21 mm; (B, G, K) DHMECN 19037 Holotype, female, SVL: 18.21 mm; (C, H, L) DHMECN 19035, female, SVL: 17.49 mm; (D, I) DHMECN 19028, female, SVL: 16.81 mm; (E). DHMECN 19031, female, SVL: 17.57 mm; (F, J, M) DHMECN 19036, female, SVL: 16.55 mm. Photographs by Christian Paucar.

-78.09140; 2,605 m), collected on 27 July 2023 by at Miguel Urgilés-Merchán & Christian Paucar Veintimilla.

**Paratypes** (Fig. S4). A total of seven males (7) with the same type locality and the same data collection. DHMECN 19028, (0.82537, -78.09639; 2,498 m); DHMECN 19029, (0.82534, -78.09631; 2,511 m); DHMECN 19031, (0.82365, -78.09532; 2,483 m); DHMECN 19032, (0.82483, -78.0962; 2,499 m); DHMECN 19033, (0.82503, -78.0962; 2,487 m); DHMECN 19034, (0.82361, -78.0949; 2,512 m); DHMECN 19035, (0.82364, -78.0946; 2,494 m) and one juvenile (1): DHMECN 19036, (0.81995, -78.09433; 2,570 m).

**Diagnosis.** *Pristimantis broaddus* sp. nov. is a member of the *Pristimantis myersi* species group and *P. celator* species clade, characterized by the following combination of characters: (1) dorsal skin shagreen with scattered low tubercles and low warts, some aggregation of

tubercles extending towards the post sacral region; flanks with subconical tubercles aggregated mainly in the axillary region; subconical tubercles on the ventral edge of the lower jaw; dorsolateral fold present well-defined visible from the scapular region to or slightly beyond the sacral region (ilium), discoidal fold present slightly visible, belly areolate with tiny scattered tubercles; (2) tympanum present, superficially visible, tympanic annulus weekly defined, round, horizontal diameter of tympanum equal to 49,23% of eye diameter ($n = 9$), antero-dorsal margin with a supratympanic fold composed by low warts, subconical postrictal tubercles present; (3) snout short, broadly rounded in dorsal view, short, curved, curved, tip of snout protruding in lateral view; (4) upper eyelid with one conical tubercle surrounded with several lower subconical tubercles; cranial crest absent; (5) dentigerous processes of vomers positioned, obliques in outline, posterior to level of choanae, minute and embedded in roof of mouth, barely visible and therefore sometimes considered to be absent; (6) males unknown; (7) finger I shorter than finger II; discs broad and expanded mainly in the fingers III–IV, with circunmarginal grooves; (8) fingers with strongly crenulated lateral fringes, mainly in Finger IV; (9) ulnar tubercles present, subconical placed on the outer edge of the ulna; (10) heel with an elongated conical tubercle surrounded by lower rounded tubercles, outer edge of the tarsus with a row of subconical tubercles, internal tarsal fold weakly defined; (11) metatarsal tubercle slightly larger than the external one which is rounded and subconical; very few supernumerary tubercles and barely visible; (12) toes with weakly defined lateral fringes, basally pronounced, interdigital membrane absent, toe V longer than toe III, does not reach the base of the distal subarticular tubercle of the IV toe; (13) Dorsal coloration back in light neutral gray to medium neutral gray (sayal brown to burnt umber in life) with speckled marks disky brown or vandyke brow (spectrum green to dark spectrum green in life). Ventrally pale neutral gray to dark neutral gray to light brown (with warm sepia, scattered reddish-brown tubercles in life). Distinctive dark brown trapezoidal-shaped mark on the throat, bordered in cream, with a green outer heel edge and gold iris; (14) males unknow, females SVL = 16.81–18.42 mm (mean = 17.62 mm, $n = 8$).

**Comparison with other species** (Figs. 2, 5, 6, 7, 8). The new species differs from the most closely related species (*P. verecundus*, *P. mutabilis*, *P. robayoi* sp. nov., *P. satheri* sp. nov., and *P. praemortuus* sp. nov.) by the presence of dorsolateral fold well-defined visible from the scapular region to or slightly beyond the sacral region (ilium) (complete dorsolateral glandular folds extending to the first quarter of the ilium in *P. satheri* sp. nov., dorsolateral fold reaching just before the sacrum in *P. verecundus*, dorsolateral fold only reach the level of sacrum in *P. mutabilis*, dorsolateral folds extending the entire length of the ilium in *P. robayoi* sp. nov.); a white belly without V-shaped markings on the throat (present in *P. verecundus*, *P. broaddus* sp. nov., *P. praemortuus* sp. nov., reduced or inconspicuous in *P. mutabilis* and *P. robayoi* sp. nov.; the dorsal surface of the tibia covered by transverse rows of glandular tissue in *P. broaddus* sp. nov., *P. verecundus* and *P. praemortuus* sp. nov. (outlining the extreme edge of the heel in *P. robayoi* sp. nov., and dorsal surface of tibia covered by glandular tissue in *P. satheri* sp. nov.); and a gold red in *P. broaddus* sp.nov. (bright orange-copper with a black edge in *P. robayoi* sp. nov., coppery gold in *P. praemortuus* sp. nov., iris cream to golden with thin black reticulation and a

reddish-brown horizontal streak in *P. mutabilis*, and wine-colored in *P. satheri* sp. nov.). Tympanic annulus weakly defined in *P. broaddus* sp. nov. (clearly defined, round, posterior border of tympanic annulus with a thick fold in *P. praemortuus* sp. nov.), fingers with weakly defined lateral fringes in *P. broaddus* sp. nov.(fingers with lateral fringes, strongly crenulated mainly in finger IV in *P. praemortuus* sp. nov.), toes with weakly defined lateral fringes, basally pronounced , interdigital membrane absent in *P. broaddus* sp. nov. (toes with weakly defined lateral fringes, basally pronounced, interdigital membrane absent, outer edge of the foot and toe IV crenulated in *P. praemortuus* sp. nov.).

Genetic distances with close-related species (*Pristimantis broaddus* sp. nov. *vs. P. praemortuus* sp. nov.) have 3.54%. The genetic distances with other species in this clade range from 3.54%–12.49% (Table S2).

**Description of the holotype** (Figs. 2, 5, 8, 10). Adult female (DHMECN 19037), head longer than wide, snout short, broadly rounded in dorsal view, curved, tip of snout protruding in lateral view, eye-nostril distance 9.82% of SVL, *canthus rostralis* slightly concave, defined, loreal region concave, nostrils protuberant, directed dorsolaterally; interorbital area slightly elevated, interorbital fold absent, interorbital distance wider than upper eyelid, 85.39%; cranial crest absent, X-shaped occipital crease (evident in life); upper eyelid with one conical tubercle surrounded with several lower subconical tubercles (reduced by preservation effects), tympanum present, superficially visible, tympanic annulus weakly defined, round, slightly visible dorsally, diameter of tympanum equals 49.23% of eye diameter, antero-dorsal margin with a supratympanic fold composed by low warts, subconical postrictal tubercles present; choanae small, rounded, not concealed by palatal shelf of maxilla; dentigerous processes of vomers oblique in outline, posterior to level of choanae, minute and embedded in roof of mouth, barely visible, two minute teeth. Tongue longer than wide, notched posteriorly, posterior 60% not adhered to mouth floor.

Texture of dorsum dorsal skin shagreen with scattered low tubercles and warts, some aggregation of tubercles extending towards the post sacral region; flanks with subconical tubercles aggregated mainly in the axillary region; subconical tubercles on the ventral edge of the lower jaw; dorsolateral fold present well-defined visible from the scapular region to or slightly beyond the sacral region (ilium), discoidal fold present slightly visible, venter areolate with tiny scattered tubercles; two evident ulnar tubercles present, subconical placed on the outer edge of the ulna; outer palmar tubercle lobular, segmented into three parts, approximately the same size of elongated outer thenar tubercle; supernumerary tubercles present; subarticular tubercles defined, rounded in lateral view. Fingers strongly crenulated, mainly in finger IV; finger I shorter than finger II; discs broad and expanded mainly in the fingers III–IV, with circunmarginal grooves, about 1.8× the size of the digit. Hind limbs slender (TL 47.02% of SVL; FL 44.6% of SVL); heel with an elongated conical tubercle surrounded by lower rounded tubercles, outer edge of the tarsus with a row of subconical tubercles, internal tarsal fold weakly defined; metatarsal tubercle slightly larger than the external one which is rounded and subconical; few barely visible supernumerary tubercles; toes with weakly defined lateral fringes, basally pronounced, interdigital membrane absent, toe V, which is longer than toe III, does not extend to the base of the distal subarticular tubercle of toe IV.

**Holotype coloration in preservative** (Fig. 10). Background color in pale neutral gray (296) and light neutral gray (297). With distinctive medium dorsal mark dark grayish brown (284). Surface of the head and occipital region with spots in shades of pinkish white (216). Dorsolateral folds pale Pinkish buff (3). Delineated markings in sepia (286) and pale neutral gray (296) on the scapular region and dorsum. Ovoid markings on the flanks, diagonal bars on the posterior surfaces of the thighs, forearms, tibiae, and feet with markings in shades of drab (19) to dusky brown (285). Interspaces of bars on posterior surfaces of thighs and forearms pale neutral gray (296). Diagonal bars drab gray (256) extend from back of head to middle of flanks; anterior half of white with diagonal bars dusky brown (285). Inguinal region in spectrum orange background (9) and ventral surfaces of tibia in light fresh color (250). Ventrally pale neutral gray (296), finely punctuated with dark neutral gray (299); chest and inverted trapezoidal mark on throat dark neutral gray (299).

**Holotype coloration in life** (Figs. 6, 11). Anterior surface of the dorsum russet color (44) with distinctive occipital, loreal, posterior surfaces of the thighs, heels, and base of the fingers IV-III, apple green (104). Posterior surface of dorsum beige (254) with dorsolateral buff folds (five), flank bars pale lime green (112), ovoid flank markings, smoke gray (267). Bars on posterior surfaces of thighs and tibia, anterior flank markings, and supratympanic stripe ground cinnamon (270). Labial stripes cinnamon drab (50). Ventrally, dark pearl gray (280) with protruding ground cinnamon colored tubercles (270). Groin and anterior surfaces of the tibia poppy red (63). Trapezoidal marl on chin and lines on throat and ventral surface of thighs Dusky brown (285). Iris sulphur yellow (80).

**Measurements (in mm) of holotype.** SVL = 18.21; HW = 6.04; HL = 6.26; ED = 1.97; IOD = 2.11; EN = 1.79; TD = 0.97; TL = 8.15; EW = 1.52; FoL = 8.88; HaL = 4.75; FW = 0.54; TW = 0.57.

**Variation.** Morphometric variation in the type series (only females) is shown in Table 2 and Fig. S3. Variations in coloration in life (Fig. 11), dorsal coloration from sayal brown (41) (DHMECN 19029, 19035) to burnt umber (48) (DHMECN 19031, 19036), with distinctive diagonal stripes on the flanks ranging from straw yellow (53) (DHMECN 19029) to pale cinnamon (54) (DHMECN 19035), bordered in cream. The dorsum with speckled marks of spectrum green (129) (DHMECN 19028, 19029, 19037) to dark spectrum green (130) (DHMECN 19036), dorsolateral folds straw yellow (53) (DHMECN 19035, 19028), pale cinnamon (54) (DHMECN 19029), or true cinnamon (260) (DHMECN 19036). Black supratympanic stripe with cream anterior border, irregular and incomplete, does not reach the postrictal tubercle. Labial stripes burnt umber (48) (DHMECN 19029, 19035) or jet black (300) (DHMECN 19036) and transversal tibial (glandular) straw yellow to pale cinnamon (54) (DHMECN 19035, 19028, 19036). Outer heel edge spectrum green (DHMECN 19035, 19028, 19036). Trapezoidal mark on chin from dusky brown (285) (DHMECN 19037), amber (51), (DHMECN 19035) to sepia (286), separated by bars, dark pearl gray (280), pales horn color (11) and olive horn color (16). Ventrally dark pearl gray (280) (DHMECN 19037), pale neutral gray (296) (DHMECN 19035) to olive brown (278) (DHMECN 19036), speckled with warm sepia (40), scattered reddish-brown tubercles.

In preservative (Fig. S4), anterior dorsal pattern varies from back in light neutral gray (297) (DHMECN 19037) to medium neutral gray (298), (DHMECN 19031–32, 19035).

Medium dorsal mark from dark grayish brown (284) (DHMECN 19037) with dusky Brown (285) (DHMECN 19031–32, 19028) and vandyke brown (281) (DHMECN 19028). Specimen DHMECN 19036 with white middorsal stripe extending from snout to the vent. Flanks with dorsolateral oblique bands and, shades of drab (19) (DHMECN 19037), beige (254) (DHMECN 19035) to vandyke brown (281) (DHMECN 19031–32). Groin and anterior surfaces of the tibia vary from poppy red (63) (DHMECN 19037), spectrum orange (nine) (DHMECN 19032) to salmon (251) (DHMECN 19031). Belly varies from pale neutral gray (296) (DHMECN 19032) with dark spots scattered to dark neutral gray (299) (DHMECN 19031), with heavy mottled reaching throat, chest, and ventral limbs. Specimen DHMECN 19036, ventral coloration extending on the waist delineating a trapezoidal mark shape jet black (300). Specimen DHMECN 19035 is the only specimen with V mark shape.

**Etymology.** The specific epithet *broaddus* is a noun in apposition in recognition of Callie Broaddus. She is an American conservationist, photographer, and filmmaker. Founder of Reserva: The Youth Land Trust, a nonprofit organization that empowers youth to conserve threatened species and habitats. Since 2019, she has worked primarily in the Tropical Andes of Ecuador, collaborating with local partners to protect the Dracula Reserve ecological corridor. Callie has a strong background in conservation and youth activism, with experiences from various international expeditions and conferences. Previously, she was a senior designer at National Geographic Kids and has been involved in multiple outreach and environmental education projects.

**Distribution and natural history.** *Pristimantis broaddus* sp. nov. is known only from the type locality in the El Pailón sector, of the Golondrinas Protective Forest of the province of Carchi, between 2,483 to 2,605 m of elevation, in the Mira River Basin located on the slope of Cerro Golondrinas (Fig. 3). This species is found in the montane evergreen forest of the Western Cordillera of the Andes (*MAE, 2013*), characterized by a semi-open canopy up to 20 m high, covered by epiphytes, orchids, bromeliads, bryophytes, and ferns, with dominance of suros (*Chusquea* sp.). The nine known specimens of *Pristimantis broaddus* sp. nov. were found in the lower layer of the forest between 10 and 110 cm, on leaves of bushes and ferns at night. *P. broaddus* are found in sympatry with the species; *Pristimantis celator*, *P. pteridophilus* complex, *P. apiculatus,* and *P. hectus.*

*Pristimantis satheri* **sp. nov.**
*Eleutherodactylus verecundus* Lynch & Burrowes, 1995. Occas. Pap. Mus. Nat. Hist. Univ. Kansas, 136: In Part: Paratype: IAvH-Am-1492.
*Pristimantis* sp.5 *Franco-Mena et al., 2023*
LSIDurn:lsid:zoobank.org:act:D756EFAA-9D18-4564-AB7F-B5E1866054E0
**Proposed standard Spanish name:** *Cutín de Justin Sather.*
**Proposed standard English name:** *Satheri's rainfrog*

**Holotype** (Figs. 2, 5–8, 12–13). DHMECN 14858, adult female, from San Jacinto de Chinambí, Jijón y Caamaño, Tulcán, Carchi province, Ecuador, (0.860927, -78.272364;

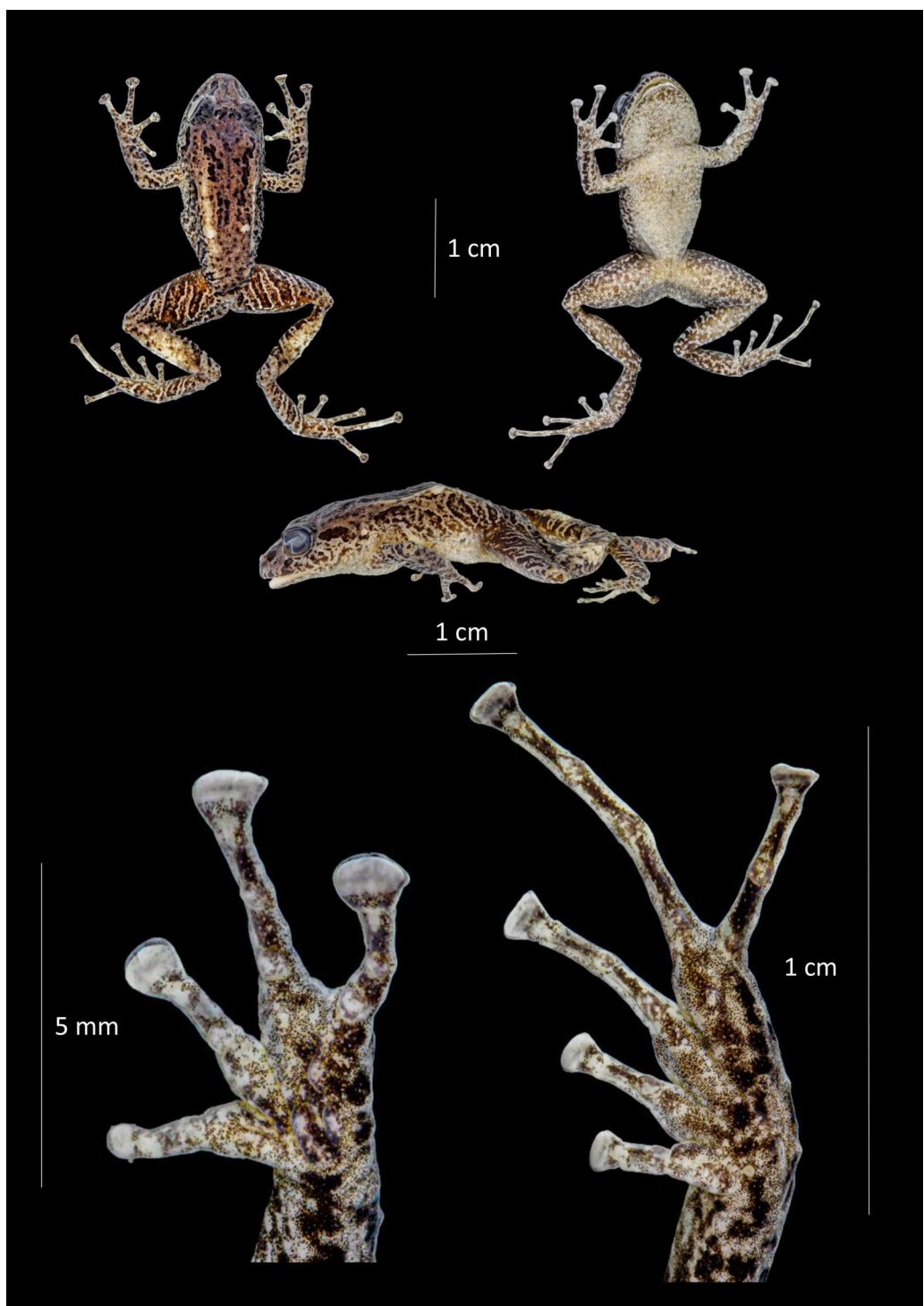

**Figure 12  Dorsal, ventral and profile views of *Pristimantis satheri* sp. nov., female adult (DHMECN 14858) and detail of hand and foot.**  Photographs by Christian Paucar.

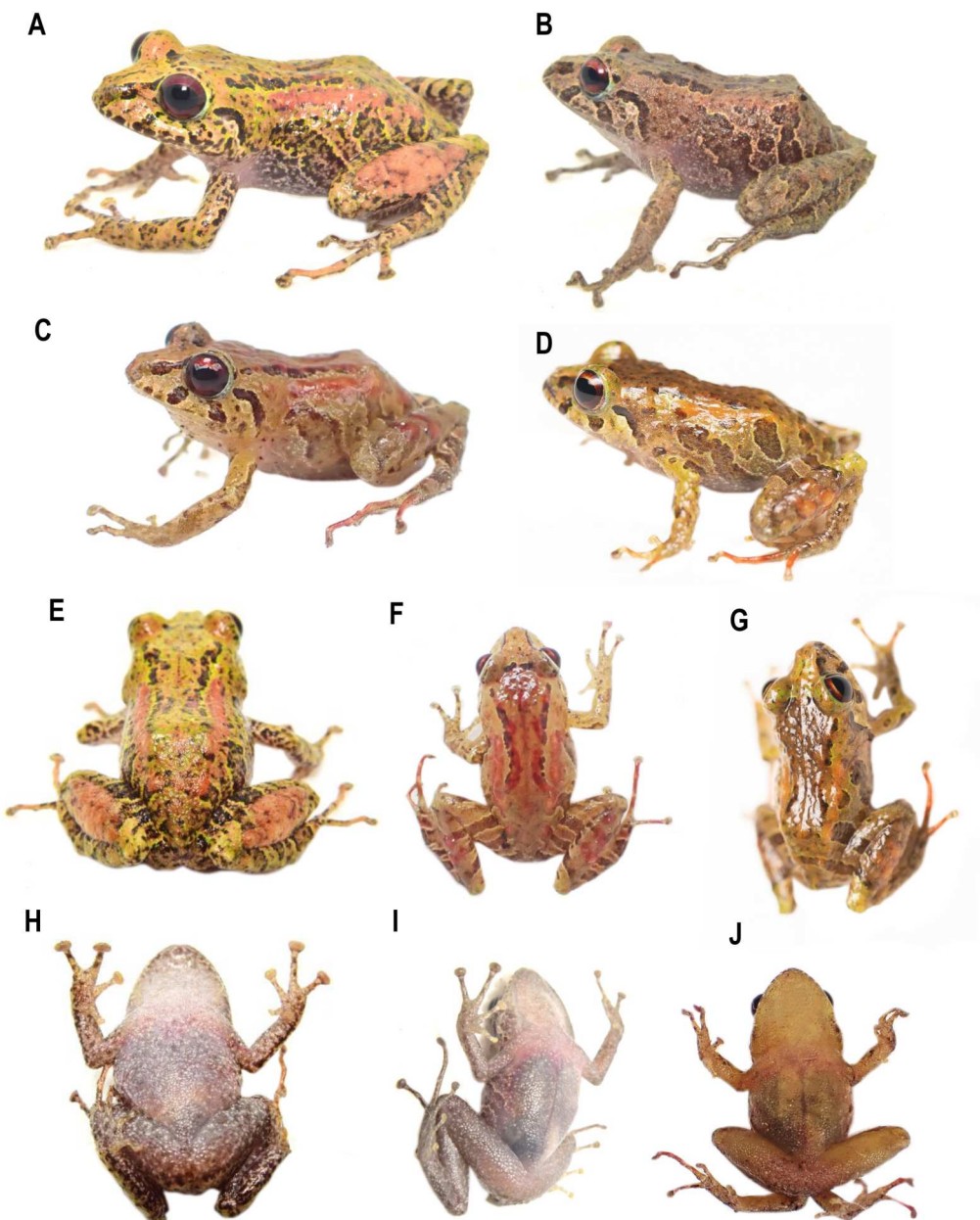

**Figure 13  Variation in life of the type series of *Pristimantis satheri* sp. nov.** (A, E, H) DHMECN 14858, Holotype, female, SVL: 23.74 mm; (B, I) DHMECN 14885, female, SVL: 22.08 mm; (C, F, J) DHMECN 19538, male, SVL: 19.79 mm; (D, G) DHMECN 19538, juvenile, SVL: 14.71 mm. Photographs by Mateo Vega-Yánez (A, B, E, H, I), Julio C. Carrión-Olmedo (D, G), Christian Paucar (C, F, J).

2,156 m), collected on 15 June 2019, by at Mario H. Yánez-Muñoz; Mateo Vega-Yánez and Daniel Padilla.

**Paratypes** (Fig. S4). A total of five specimens (5). Adult females (one): DHMECN 14885, with the same type locality and data collectors, (0.861354, -78.276026; 2,196 m.), collected on 17 June 2019; Adult males (two): DHMECN 16578, from Bloque 18, Dracula

Reserve, Tulcán, Carchi province, Ecuador, (0.87830 -78.20604; 2,141 m), collected on 15 October 2021 by at Ross Mayner & Jaime Culebras; DHMECN 19538, from the Bloque 20, Dracula Reserve, Tulcán, Carchi province, Ecuador, (0.83566, -78.22842; 2,394 m), collected on 5 December 2023 by at Mario H. Yánez-Muñoz; Christian Paucar Carlos Ríos & Miguel Urgilés-Merchán; Juveniles (two): DHMECN 16574, with the same data of DHMECN16578; DHMECN 17903, Cerro Negro, Dracula Reserve, Carchi province, Ecuador, (0.8826, -78.19587; 1,742 m), collected on 5 August 2022 by at Gabriela Lagla-Chimba, Julio César Carrión-Olmedo & Milton Cantincuz.

**Diagnosis.** *Pristimantis satheri* sp. nov. is a member of the *Pristimantis myersi* species group and *P. celator* species clade, characterized by the following combination of characters: (1) dorsal skin shagreen with flanks with rounded warts, venter areolate, discoidal fold present and visible posteriorly, dorsolateral glandular folds complete, from the scapular region to the first quarter of the ilium; (2) tympanum present, superficially visible, tympanic ring well defined, round, horizontal diameter of tympanum equal to 32.75% of eye diameter ($n = 7$) and without conical tubercles; (3) snout short, subacuminate in dorsal view, rounded in profile with slightly flared lips; (4) upper eyelid with several small rounded tubercles; no cranial crest; (5) dentigerous processes of the vomer present, triangular in outline each process with 4–5 teeth; (6) vocal slits present; with nuptial pad and low vocal sac; (7) finger I shorter than finger II; broad, expanded, disks dilated with circunmarginal grooves; (8) fingers with thin lateral fringes, strongly crenulated on external edge of finger IV - extends and attenuates to finger III; (9) ulnar tubercles present, with 3–4 rounded tubercules; (10) heel with one subconical tubercle; outer edge with two subconical tubercles, dorsal surface of the tibia covered by glandular tissue; (11) small inner metatarsal tubercle oval 2 times larger than small rounded outer metatarsal tubercle; supernumerary tubercles lows; (12) toes with thin cutaneous lateral fringes, interdigital membrane absent, toe V longer than toe III, fringes developed and thick on the fingers, edge of toe V crenulated; (13) dorsally, from light green with a variegated pattern of black, with distinctive orange dorsolateral folds and tibial gland, to grayish brown; white belly without distinctive V-shaped markings; (14) small adult males, SVL = 19.73–19.79 mm (mean = 19.76, $n = 2$), females SVL = 22.08–23.74 mm (mean = 22.91 mm, $n = 2$), Juveniles, SVL = 14.7–14.71 mm (mean = 14.71 mm, $n = 2$).

**Comparison with other species** (Figs. 2, 5, 6, 7 and 8)**.** The new species differs from other species of the clade (*P. verecundus*, *P. mutabilis*, *P. robayoi* sp. nov., *P. broaddus* sp. nov., and *P. praemortuus* sp. nov.) by the presence of complete dorsolateral glandular folds extending to the first quarter of the ilium (partial dorsolateral folds reaching just before the sacrum in *P. verecundus*, dorsolateral folds reaching the sacrum in *P. broaddus* sp. nov. and *P. praemortuus* sp. nov., only reach the level of sacrum in *P. mutabilis*, dorsolateral folds extending the entire length of the ilium in *P. robayoi* sp. nov.; a white belly without V-shaped markings on the throat (present in *P. verecundus*, *P. broaddus* sp. nov., *P. praemortuus* sp. nov., reduced or inconspicuous in *P. mutabilis* and *P. robayoi* sp. nov.); and the dorsal surface of the tibia covered by glandular tissue (outlining the extreme edge of the heel in *P. robayoi* sp. nov., and in transverse rows on the tibia in *P. verecundus*, *P. broaddus* sp.nov., and *P. praemortuus* sp. nov.); and a red wine-colored iris (bright orange-copper

with a black edge in *P. robayoi* sp. nov. , coppery gold in *P. praemortuus* sp. nov., gold in *P. broaddus* sp. nov., iris cream to golden with thin black reticulation and a reddish-brown horizontal streak in *P. mutabilis*).

Genetic distances with close-related species (*Pristimantis satheri* sp. nov. *vs. P.* sp. 4.) have 11.14%. The genetic distances with other species in this clade range from 10%–14% (Table S2).

**Description of the holotype** (Figs. 12 and 13). Adult female 23.7 mm SVL. Head longer than wide, snout short, subacuminate in dorsal and lateral view, eye-nostril distance 84.39% of horizontal eye diameter. Canthus rostralis straight, defined, loreal region concave, nostrils directed slightly posterior laterally; interorbital area flat, no interorbital fold, interorbital distance wider than upper eyelid, 76.33%; no cranial crest. Upper eyelid bearing small rounded tubercles, tympanum present, tympanic membrane not differentiated from surrounding skin, tympanic ring slightly evident bellow skin, upper margin covered by low tympanic fold, tympanum visible dorsally, diameter of tympanum equals 33.75% of eye diameter, underside of tympanum with three enlarged rounded tubercles; choanae small, oval in outline, not covered by palatal floor of maxilla; dentigerous processes of vomers present, triangular oblique in outline with four to five teeth, tongue longer than broad as broad as long, oval in shape 40% attached to floor of mouth.

Texture of dorsum finely granular with a pair of small rounded tubercles in occipital area, complete dorsolateral glandular folds with some rounded tubercles along the folds, flanks strongly rough with small rounded warts widespread on the flanks; belly strongly areolate; discoidal fold weakly evident, cloaca with finely granular texture. Slender arms with thick glandular skin, rounded tubercles on dorsal and ventral surface of forearm. Ulnar tubercles present; small subconic and rounded; broad truncated disks on fingers II to IV, finger round slightly expanded, all fingers bearing circunmarginal grooves, subarticular tubercles rounded and flattened in lateral view, fingers with thin lateral cutaneous ridges; thenar tubercle oval enlarged with some irregular borders almost half size of palmar "horseshoe" shape tubercle, palmar surface with low flat and inconspicuous supernumerary tubercles; hind limbs slender, length of tibia equals 49.03% of SVL, glandular patch on outer surface of tibia; enlarged rounded tubercles on the heel surrounded by small tubercles, row of rounded tubercles along outer edge of the tarsus, inner tarsal fold absent, toes with thin cutaneous ridges, toes I to V widely expanded bearing circunmarginal grooves without digital membranes; subarticular tubercles rounded and flattened in profile view; expanded disks on all toes, larger than those of the hand. Toe V longer than III, not extending beyond subarticular tubercle of toe IV.

**Holotype coloration in preservative** (Fig. 12). Dorsal surface, forelimbs, and hindlimbs ferruginous (35) background, with a thin sepia (279) interorbital, dorsal, forelimbs, and hind limb with bands. Dorsolateral glandular folds and glandular folds light yellow ocher (13). Flanks and dorsal surface of hindlimbs olive brown bands (278), finely outlined in white with interspaces clay color (20). Smoky white (261) belly, finely dotted with smoke gray (267). Ventral surface of thighs and legs, variegated with smoke gray (267). Sepia (279) canthal, supralabial and supratympanic markings.

**Holotype coloration in life** (Fig. 13). Dorsal coloration on a light buff background (2), with distinctive marks and light pistachio outlines (101). Dorsolateral glandular folds and glandular folds flame scarlet (73). Belly and throat pale neutral gray (297), with cream spots. Dorsal markings, supratympanic lines, labial bars, flank bars, front and hind limbs jet black (300). Gem Ruby (65) colored iris.

**Measurements (in mm) of holotype.** SVL = 23.74; HW = 8.51; HL = 9.23; ED = 3.14; IOD = 2.62; EN = 2.65; TD = 1.06; TL = 11.63; EW = 2; FL = 11.86; HaL = 7.13; FW = 1.27; TW= 1.04.

**Variation.** *Pristimantis satheri* sp. nov. shows sexual dimorphism in body size, females 0.86 times larger than males (Fig. S3). Other morphometric variations of the type series are showing in Table 2.

In life (Fig. 13), dorsal background coloration ranges from light buff (2) with pistachio markings (101) (DHMECN 14858) to cinnamon drab (21) with drab-gray markings (256) (DHMECN 14885). The dorsal glands are observed to exhibit a range of colors, including flame scarlet (73) (DHMECN 14858), medium chrome orange (75) (DHMECN 17903), and drab-gray (256) (DHMECN 14885). Dorsal end bars are jet black (300) or in lighter shades (DHMECN 14858), or they may be true cinnamon (260) (DHMECN 17903). The ventral surface is characterized by a neutral gray background (297) (DHMECN 14858, 14885) with cream-colored spots or a uniform cream color (12) (DHMECN 17903).

In preservative (Fig. S5), dorsally the background color varies from ferruginous (35) (DHMECN 14858), mahogany red (34) (DHMECN 16578) to true cinnamon (260) (IAvH-Am-1492). Bars on the posterior surfaces of the thighs sepia (279) (DHMECN 16578), russet (44) (DHMECN 19538), true cinnamon (260) (IAvH-Am-1492).

**Etymology.** The specific name *satheri* is a noun in the genitive case and is a patronym Justin Sather. Justin, a 13-year-old American male, is an ardent environmentalist. Justin Sather was inspired to found the Justin's Frog Project by a dual concern: his love for frogs and the decline of their populations due to pollution and habitat destruction. He has safeguarded over 100 acres of rainforest in Ecuador, removed 1,000 pounds of waste from polluted rivers, and planted 500 trees. Justin's objective is to foster awareness and prompt action to safeguard the planet.

**Distribution and natural history.** *Pristimantis satheri* sp. nov.is known in the type locality in the San Jacinto de Chinambi sector, block 18 and block 20 of the Dracula Reserve, in the province of Carchi, between 1,742 and 2,394 m elevation in Ecuador and Reserva La Planada, Colombia, in the Mira River Basin (Fig. 3). This species is found in the evergreen montane forest of the Western Cordillera of the Andes (*MAE, 2013*), characterized by an open canopy with trees up to 20 m high, covered by epiphytes, orchids, bromeliads, bryophytes, and ferns. The six known specimens of *Pristimantis satheri* sp. nov.were found active at night perching on leaves of shrubs and bromeliads in the lower and middle stratum of the forest, between 100 to 180 cm in height. They were found in sympatry with: *P. pteridophilus* complex, *P. apiculatus*, *P. hectus*, *P.* grp. *devillei*, and *P. praemortuus* sp. nov. (Fig. 3).

*Pristimantis robayoi* **sp. nov.**

*Eleutherodactylus verecundus* Lynch & Burrowes, 1995. Occas. Pap. Mus. Nat. Hist. Univ. Kansas, 136: In Part: Partype: IAvH-Am-1801.
***Common name in Spanish:*** *Cutín de Robayo*
***Suggested common English name:*** *Robayo's rainfrog.*

**Holotype** (Figs. 2, 6–8, 14–15). DHMECN 17894, adult female, from the Dracula Reserve, Chical, Tulcán, Carchi province, Ecuador, (0.86771, -78.19961; 2070 m), collected on 2 August 2022 by Gabriela Lagla-Chimba, Julio Cesar Carrión & Milton Cantincuz.

**Paratypes** (Fig. S3). A total of fourteen specimens of the type series. Adult females (7): DHMECN 14884, from San Jacinto de Chinambí, Jijón y Caamaño, Tulcán, Carchi province, Ecuador, (0.860927, -78.272364; 2,156 m), collected on 15 June 2019, by at Mario H. Yánez-Muñoz; Mateo Vega-Yánez and Daniel Padilla.; DHMECN 14947, DHMECN 14948 and DHMECN 14960 from sendero al río Cumbe, Maldonado, Tulcán, Carchi province, Ecuador, (0.89704, -78.11841; 2,012 m), collected on 24 October 2019 by at Miguel Andrés Urgilés Merchán, Mateo Vega Yánez, Cristian Daniel Agila & Rafael Mena; DHMECN 16573, from the Cerro Negro, Dracula Reserve, Chical, Tulcán, Carchi province, Ecuador, (0.87663, -78.20535; 2,160 m), collected on 15 August 2021 by Ross Maynard; DHMECN 19429, from the Dracula Reserve, Cerro Golondrinas, Chical, Tulcán, Carchi province, Ecuador, (0,86623, -78,20392; 2,227m), collected on 16 September 2023 by Juan Pablo Reyes-Piug. Adult males (3): DHMECN 14979, with the same data of DHMECN 14960; DHMENC 16150, from the La Esperanza, Quinshul, Tulcán, Carchi province, Ecuador, (0.92704, -78.24178; 1,979 m); collected on 27 March 2021 by Mario H. Yánez-Muñoz & Juan P. Reyes; DHMECN 16567 with same data of 14960. Juveniles (4): DHMECN 14886, from the Rio Chinanbi, San Jacinto de Chinanbi, Mira, Carchi province, Ecuador, (0.861354, -78.275964; 2,193 m), collected on 17 June 2019 by at Mario Humberto Yánez Muñoz, Mateo Vega Yánez & Daniel Padilla; DHMECN 14950, from the Dracula Reserve, Cerro Golondrinas, Chical, Tulcán, Carchi province, Ecuador (0.89704, -78.11841; 2,012 m), collected on 16 September 2023 by Juan Pablo Reyes-Piug; DHMECN 14953, (0.89704, -78.11841; 1,791 m), from Sendero al río Cumbe, Maldonado, Tulcán, Carchi province, Ecuador, collected on 24 October 2019 by at Miguel Andrés Urgilés Merchán, Mateo Vega Yánez, Cristian Daniel Agila & Rafael Mena; DHMECN 16575, from the Dracula Reserve, Chical, Tulcán, Carchi province, Ecuador, (0.87635, -78.20551; 2,161 m), collected on 15 August 2021 by Ross Maynard.

**Diagnosis.** *Pristimantis robayoi sp. nov.* is a member of the *Pristimantis myersi* species group and *P. celator* species clade, characterized by the following combination of characters: (1) complete dorsolateral folds, extending below occipital region through the scapula to the ilium, with a pair of occipital and scapular short sub-conical tubercles, flanks with small, low warts aligned dorsolaterally, flattened warts, areolate belly, discoidal fold present and visible posteriorly, dorsolateral folds finely defined; (2) tympanum hidden below the skin, tympanic annulus present, round, well-defined, horizontal diameter of the tympanum equal to 32.05% of the eye diameter ($n = 14$), postrictal sub-conical tubercles present; (3) snout short, rounded in dorsal and profile views; (4) upper eyelid with several low sub-conical

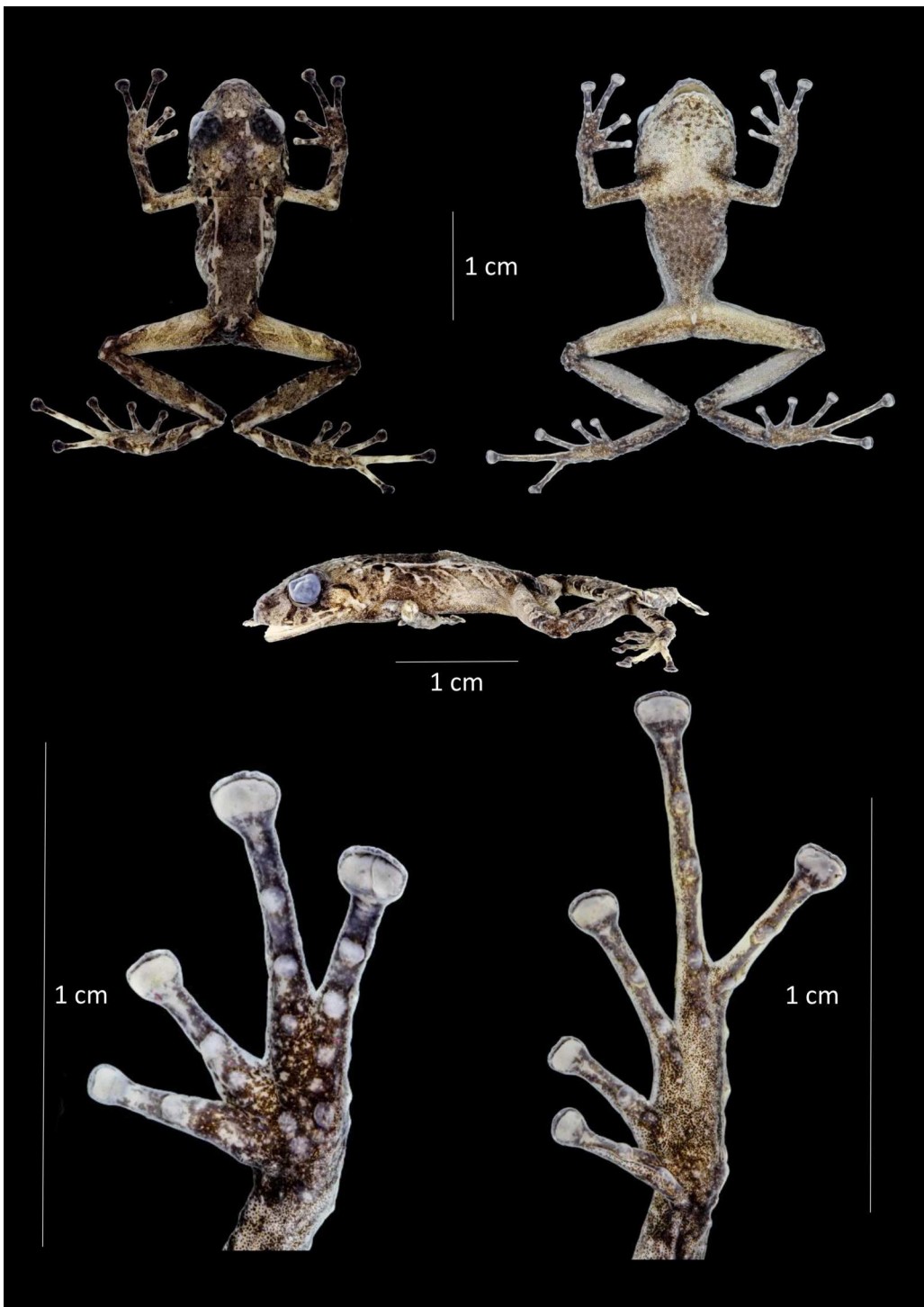

**Figure 14  Dorsal, ventral and profile views of *Pristimantis robayoi* sp. nov., male adult (DHMECN 17894) and detail of hand and foot.** Photographs by Christian Paucar.

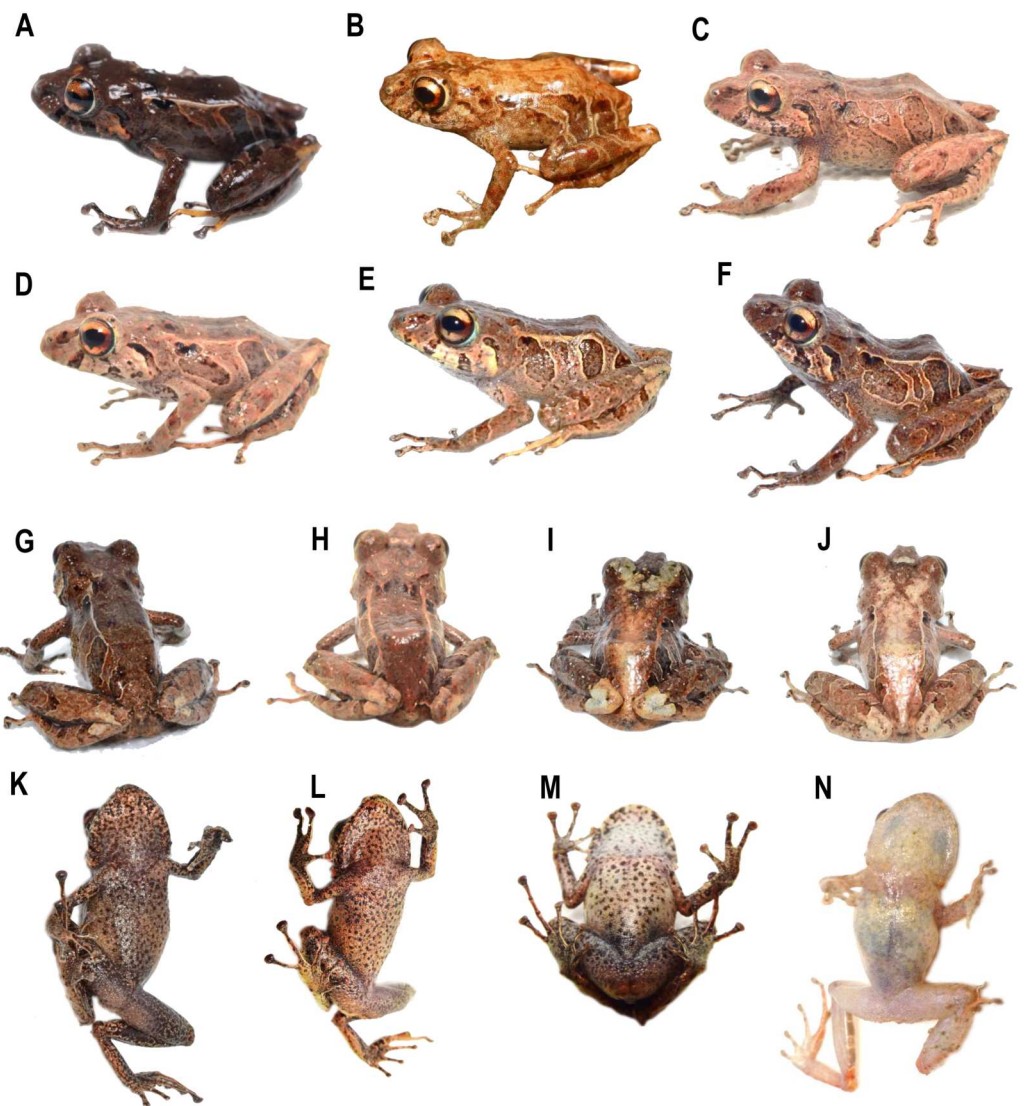

**Figure 15** **Variation in life of the type series of *Pristimantis robayoi* sp. nov.** (A) DHMECN 17894 Holotype, male, SVL: 24.22 mm; (B, N) DHMECN 16150, female, SVL: 21.01 mm; (C, L) DHMECN 14884, female, SVL: 29.05 mm; (D, H, M) DHMECN 14885, male, SVL: 20.15 mm; (E) DHMECN 14953, juvenile, SVL: 20,15 mm; (F) DHMECN 14947, female, SVL: 25,83 mm; (G, K) DHMECN 14960, female, LRC: 27,96; (I) DHMECN 14950, juvenile, SVL: 18.16 mm; (J) DHMECN 14948, female, SVL: 25.18 mm. Photographs by Mateo Vega-Yánez (A, C–M, E, H, I), Juan P. Reyes-Puig (B, N).

tubercles and one small conical tubercle, small rounded interobital tubercle; no cranial crest; (5) vomerine odontophores present, triangular in outline, each odontophore with 7–9 teeth; (6) vocal slits not visible; nuptial pad absent, no vocal sac; (7) finger I shorter than Finger II; expanded discs wider than the digits, fingers with circumferential grooves and lateral fringes; (8) sub-conical ulnar tubercles present; (9) heel with a sub-conical tubercle surrounded by lower ones, three tubercles on the edge of the tarsus, glands present outlining the outer edge of the heel; (10) two metatarsal tubercles, inner metatarsal

tubercle oval, three times the size of the outer sub-conical one, low supernumerary tubercles present; (11) toes with lateral fringes that widen basally, no interdigital membrane; (12) toes with thin lateral fringes without digital membranes, toe V longer than toe III; (13) dorsally dark brown with distinctive black scapular marks, dorsolateral folds, diagonal bands on the flanks, postrictal area, and glandular surface of the outer edge of the heel orange-brown, diagonal labial bars, black supratympanic fold, belly light brown to dark brown densely speckled, iris bright orange-copper with a black edge; (14) adult males, SVL= 12.46–17.04 mm (mean = 20.7, $n = 3$), females SVL= 21.78–29.05 mm (mean = 25.42 mm, $n = 7$).

**Comparison with other species** (Figs. 2, 5, 6, 7, 8). The new species differs from its closest nominal lineages (*P. verecundus, P. mutabilis, P. satheri* sp. nov., *P. broaddus* sp. nov., *P. praemortuus* sp. nov.), by complete dorsolateral folds, extending below occipital región through the scapula to the illium (complete dorsolateral glandular folds extending to the first quarter of the ilium in *P. satheri*, dorsolateral fold reaching just before the sacrum in *P. verecundus*, dorsolateral fold only reach the level of sacrum in *P. mutabilis*, dorsal fold present well-defined visible from the scapular region to or slightly beyond the sacral region (ilium) in *P. boraddus* sp. nov.)*;* a with belly areolate (belly without V-shaped markings on the throat in *P. praemortuus* sp. nov., *P. verecundus, P. broaddus* sp. nov., reduced or inconspicuos in *P. mutabilis* and *P. satheri* sp. nov.); glands present outlining the outer edge of the heel (dorsal surface of tibia covered by glandular tissue in *P. satheri* sp. nov., dorsal surface of the tibia covered by transverse rows of glandular tissue in *P. broaddus* sp. nov., *P. verecundus* and *P. praemortuus* sp. nov.); iris bright orange-copper with a black edge (gold red in *P. broaddus* sp. nov., coppery gold in *P. praemortuus* sp. nov., iris cream to golden with thin black reticulation and a reddish-brow horizontal streak in *P. mutabilis*, and wine-colored in *P. satheri* sp. nov.); Tympanum hidden below the skin, tympanic annulus present, round, well-defined (tympanum present, superficially visible, tympanic ring well defined, round in *P. satheri* sp. nov., tympanic annulus clearly defined, round, posterior border of tympanic annulus with a thick fold in *P. praemortuus* sp. nov., tympanic annulus weakly defined in *P. broaddus* sp. nov.). Fingers with skin ridges that widen mainly at the base of fingers I-II and II-III, absent interdigital membrane (toes with thin cutaneous lateral fringes, interdigital membrane absent, toe V longer than toe III, fringes developed and thick on the fingers, edge of toe V crenulated in *P. satheri* sp. nov., fingers with lateral fringes, strongly crenulated mainly in finger IV in *P. praemortuus* sp. nov., fingers with weakly defined lateral fringes in *P. broaddus* sp. nov.).

Genetic distances with close-related species (*Pristimantis robayoi* sp. nov. *vs. P. verecundus*) have 8.81%. The genetic distances with other species in this clade range from 8.08%–14.79% (Table S2).

**Description of the holotype** (Fig. 14). Adult male (DHMECN 17894), head longer than wide. Snout short, rounded slightly elongate in dorsal view, rounded in profile; eye-nostril distance 10.08% of SVL, *canthus rostralis* is weakly concave in dorsal view, whereas in cross section is rounded, loreal region concave, slightly protuberant narines directed laterally; interorbital area flat, no interorbital fold, interorbital distance wider than upper eyelid, 63.10%; no cranial crest; upper eyelid with small subconic tubercles,

small rounded interorbital tubercle; tympanum and tympanic annulus evident, tympanic membrane differentiated from surrounding skin, tympanic, not visible in dorsal view; low supra tympanic fold extending from postocular to the level of the insertion of the arm; diameter of tympanum equals 33.13% of eye diameter, underside of tympanum with low rounded postrictal tubercles; choanae small, rounded in outline, not covered by palatal floor of maxilla; dentigerous processes of vomers present oblique in outline no teeth evident, tongue as broad as long, oval in shape 40% attached to floor of mouth.

Texture of dorsum is finely granular with low "W" shape fold on the occipital region, the fold is not evident on preserved specimen, a pair of small subconic tubercles in occipital and scapular areas, low dermal ridges extending posterior to the supratympanic and occipital folds, connecting to the complete dorsolateral fold, pairs of low rounded warts are present along dorsolateral folds; flanks finaly granular with low rounded tubercles, belly areolate with rounded tubercles, throat granular with low rounded tubercles; discoidal fold not evident; cloaca with granular texture; slender arms with low rounded tubercles on dorsal surface of forearm. Low rounded ulnar tubercles; broad truncated disks on fingers II to IV, rounded and slightly expanded on finger I, disks with circummarginal grooves, subarticular tubercles rounded and flattened in lateral view, all finger with thin lateral fringes; thenar tubercle oval and widely elongated, "V" irregular shape palmar tubercle, palmar surface with scarce low rounded small supernumerary tubercles, hind limbs slender, length of tibia equals 53.09% of SVL, no tubercle on outer edge of tibia, small subconical tubercle present on the heel, low inner tarsal fold present; toes with thin lateral fringes without digital membranes; subarticular tubercles rounded and flattened in profile view; expanded truncated disks on all toes except on toe I is oval. Toe V longer than III, not reaching distal subarticular tubercle of toe IV.

**Holotype coloration in preserved** (Fig. 14)**.** Dorsal surfaces are heavily pigmented with dark neutral gray (299) small punctuations, limbs bears dull banded pattern alternated light neutral gray (297) with brownish olive (292), light neutral gray; pale neutral gray (296) light irregular interorbital mark, dark dusty brown (285) suborbital and supratympanic stripes present, small black marks surrounding occipital and scapular tubercles, light cream lines extending along dorsolateral folds; flanks with oblique banded pattern altering dark and gray tones, cloacal ornamentation composed by a dark gray trapezium mark delineated by white lines; fingers, toes and digital pads dull black. Ventral coloration of abdomen ends belly dark neutral gray (299), same as ventral limbs, palm, and soles; throat and chest white light pale coloration with dark spots on the chin; ventral shanks white pale coloration.

**Holotype coloration in life** (Fig. 15)**.** Dorsal surfaces including forelimbs and hindlimbs predominant crimson (62), with antique brown (24), pale medium orange chrome 75 diffuse marks on interorbital area, dusky brown 285 on subocular and supratympanic stripes; pale medium orange chrome 75 along dorsolateral folds, heels and interspaces of banded pattern in the flanks that is mixed with chestnut 30, black marks surrounding occipital and scapular areas, iris golden with small black points and reticulations; ventral surfaces of body and limbs dark brown burnt amber (color 48), throat mottled with amber (Color 51).

**Measurements (in mm) of holotype.** SVL = 24.22; HW = 8.57; HL = 9.51; ED = 3.38; IOD = 2.9; EN = 2.44; TD = 1.12; TL = 12.86; EW = 1.83; FoL = 11.7; HaL = 7.58; FW = 1.3; TW= 1.24.

**Variation.** *Pristimantis robayoi* sp. nov. shows sexual dimorphism in body size (Fig. S3), with females 0.8 times larger than males. Other morphometric variations of the type series are shown in Table 2.

In life (Fig. 15), the dorsal pattern of coloration on a brown background exhibits a range of variations, including warm sepia (40) (DHMECN 17894), kingfisher red-brown (28) (DHMECN 14947, 14960), and orange-brown tones such as salmon (58). (DHMECN 14884), flesch ocher (57) (DHMECN 14948), chamois (84) (DHMECN 16150), and flame scarlet (73) (DHMECN 14885). The posterior flank spaces are enclosed by transverse bands that exhibit the same variation as the dorsum. Specimen DHMECN 14950 exhibits smoky white markings (262) between the posterior part of the eyelids and the skull. Additionally, specimen DHMECN 14948 displays cream color shades (12) in the mid-dorsal area, extending from the nape to the cloacal region. The ventral background coloration exhibits variation, from light yellow ocher (13) on the chest, throat, and chin, to cinnamon (21) without dots (DHMECN 14885), to cinnamon drab (21) (DHMECN14960), flesh ocher (57) (DHMECN14884) with dark melanophores and sepia-colored dots that are uniformly distributed on the ventral surface, including the throat and limbs.

In preserved specimens (Fig. S6), the dorsal surface exhibits a variation in coloration, with dark melanophores that are jet black (300) (DHMECN 19429) or dark neutral gray (299) (DHMECN 16575). Additionally, the coloration of the surface of the dorsum exhibits a medium neutral gray (298) (DHMECN 14960), progressing to dark brown tones such as olive brown (278) (IAcH1801) or true cinnamon (260) (DHMECN 14979).

**Etymology.** The specific name *robayoi* is a noun in the genitive case and is a patronym for F. Javier Robayo. Javier is the president and founder of Fundación EcoMinga in Ecuador, a nonprofit dedicated to conserving the country's rich biodiversity. He has played a pivotal role in protecting over 27,000 acres of critical habitats in the Chocó and Tropical Andes. As a biologist and educator, Robayo has led more than 200 research and teaching expeditions, contributing significantly to scientific discoveries, including new species of orchids, frogs, and rodents. His dedication extends to educating young researchers and fostering the next generation of conservationists.

**Distribution and natural history.** *Pristimantis robayoi* sp. nov., is known from three type localities: San Jacinto de Chinambi, Chical, and Maldonado in the province of Carchi, between 1,791 and 2,193 m elevation in Ecuador, and the Reserva La Planada Colombia, in the Mira River basin (Fig. 3). This species is found in the evergreen montane forest of the Western Cordillera of the Andes (*MAE, 2013*), characterized by an open canopy with trees up to 20 m high, covered by epiphytes, orchids, bromeliads, bryophytes, and ferns. The 14 known specimens of *Pristimantis robayoi*, were found active at night perching on leaves of shrubs, bromeliads, and ferns in the lower stratum of the forest, between 60 to 160 cm high. They were found in sympatry with: *P. apiculatus, P. hectus, P. actites,* and *P. quinquagesimus.*

## DISCUSSION

### The polymorphic cryptic Type Series of *Eleutherodactylus verecundus* Lynch & Burrowes, 1990

The original description was not strictly based on the holotype but included seven specimens (the holotype and six paratypes), providing a broad diagnosis of the interspecific variation within the species without detailing patterns of variation in life. During the revision of *Pristimantis* material from Colombia in the amphibian collection at the Instituto de Ciencias Naturales of the Universidad Nacional de Colombia (ICN), one of the original authors of *P. verecundus* (John D. Lynch, personal communication) had previously noted concerns regarding the lack of individual information and completeness of the data collected at the type locality, Reserva La Planada. This limitation affected the understanding of inter- and intraspecific variation within the type series, complicating subsequent revisions in the compendium of western Ecuador (*Lynch & Duellman, 1997*), which considers *P. verecundus* as a widely distributed species from northwestern central Ecuador to extreme southwestern Colombia, but later contradicted with the description of *Pristimantis mutabilis Guayasamin et al., 2015*, actual phylogenies (*Franco-Mena et al., 2023*) and present work.

Our study, which involved delimiting species from Ecuador and examining the type material deposited at the IAvH in Colombia, confirms that the type series includes two of our four new species described in this work. Specifically, paratype IAvH-AM-1492 corresponds to the taxonomic identity of *Pristimantis satheri* sp. nov. characterized by the presence of rough glandular patches on dorsal surfaces, more rounded and smooth snout; while IAvH-AM-1493 belongs to *P. robayoi* sp. nov., diagnosed by complete dorsolateral fold extending to the ilium, rather than the holotype of *P. verecundus* with more acuminated and tuberculated snout and incomplete dorsolateral folds reaching below the sacrum.

For a better understanding of species limits and intraspecific variation, a redescription of *Pristimantis verecundus* is provided as Supplemental Information 1, and as Appendix 2, a species identification key to its phylogenetically related species.

### Phylogenetic hypotheses related to the *Pristimantis celator* species group

*Lynch & Duellman (1997)* originally assigned *Pristimantis celator* and *P. verecundus* to the *P. unistrigatus* species group, based on the Toe Condition type C (fifth toe much longer than third). This is in contrast with Toe Condition type B (fifth toe longer than third but not extending to distal subarticular tubercule of fourth toe) in the *Pristimantis myersi* group.

Subsequently, *Hedges, Duellman & Heinicke (2008)* and *Padial, Grant & Frost. (2014)* employed *sensu stricto* sequences of *Pristimantis celator* and *sensu lato* sequences of *P. verecundus* (see phylogenetic comments on the *Pristimantis celator* clade) to ascertain the proximity between these species and the group of *Pristimantis myersi* species. Nevertheless, *Hedges, Duellman & Heinicke (2008)* maintained their classification as species of the *unistrigatus* species group. Subsequently, *Padial, Grant & Frost. (2014)* excluded them from the proposed groups until the publication of their findings.

*Guayasamin et al. (2015)* conducted a study to determine the species limits between *Pristimantis mutabilis* and *P. verecundus sensu lato*. They assigned these species to the *Pristimantis myersi* species group and identified high support for their phylogenetic hypothesis. Subsequently, *Franco-Mena et al. (2023)* consolidated the *Pristimantis verecundus* clade based on extensive phylogenetic sampling and determined high cryptic diversity, associated with polytomy with *Pristimantis jubatus* and the species of the *Pristimantis myersi* group (see phylogenetic comments on the *Pristimantis celator* clade).

Our hypothesis strengthens the taxonomic and phylogenetic understanding of the *Pristimantis celator* clade. Similar to *Franco-Mena et al. (2023)*, we found high cryptic diversity and consolidated *sensu stricto* sequences of *Pristimantis verecundus* in agreement with the morphological features of the holotype. In contrast, we reorganized the clade and placed it within the *Pristimantis myersi* species group (*P. myersi* subclade + *P. celator* clade + *P. jubatus* clade) and the *Trachyphrynus* subgenus (see phylogenetic comments on the *Pristimantis celator* clade). According to the protocol of *Vences et al. (2013)* and to other global trends of maintaining taxonomic stability of hyperdiverse groups of anurans (*e.g.*, *Mahony et al., 2024*), we consider that our phylogenetic hypothesis is consistent with previous work (*Hedges, Duellman & Heinicke, 2008*; *Rivera-Correa & Daza, 2016*; *González-Durán et al., 2017*; *Jetz & Pyron, 2018*; *Bejarano-Muñoz et al., 2022*) supporting the *Trachyphrynus* clade as a monophyletic group equivalent to a subgenus composed of four groups of species (*P. myersi* + *P. devillei* + *P. letolophus* + *P. boulengeri*).

In line with recent phylogenetic studies, identifying morphological synapomorphies within the groups or subgenera of *Pristimantis* cannot be used as a sole diagnostic character (*González-Durán et al., 2017*; *Ospina-Sarria & Duellman, 2019*; *Ron, Merino & Ortiz, 2024*; *Bejarano-Muñoz et al., 2022*). Additionally, the four groups that constitute the subgenus *Trachyphrynus* may exhibit homoplastic characters across the *Pristimantis* genus. These include cranial crests (*devillei* group), iridophores (*boulengeri* group), hyperdistal tubercles (*leptolophus* group), and finger conditions B and C (two conditions in the *myersi* group), all found within various monophyletic groups of *Pristimantis* (*Ospina-Sarria & Duellman, 2019*; *Ron, Merino & Ortiz, 2024*; *Bejarano-Muñoz et al., 2022*).

## Hypothesis of speciation and radiation of the *Pristimantis celator* clade

The role of mountain systems, such as the Andes, in driving species diversification is further corroborated by studies on the geological and evolutionary processes shaping mountain biodiversity (*Antonelli et al., 2018*; *Perrigo, Hoorn & Antonelli, 2019*). As detailed by *Perrigo, Hoorn & Antonelli (2019)*, the orogeny of mountain ranges, including the Andes, has created complex landscapes that serve as hotspots for both allopatric and sympatric speciation.

The phylogenetic analysis of the *Pristimantis celator* clade (14 species) reveals a distinct radiation confined to the western slopes of Ecuador and the southernmost region of Colombia. Specifically, 64% of the clade (nine species) inhabit montane forests at elevations ranging from 1,400 m to 2,300 m in the binational basin of the Mira River. In contrast, 36% (six species) occupy similar elevations, including the Andean foothills up to 600 m in

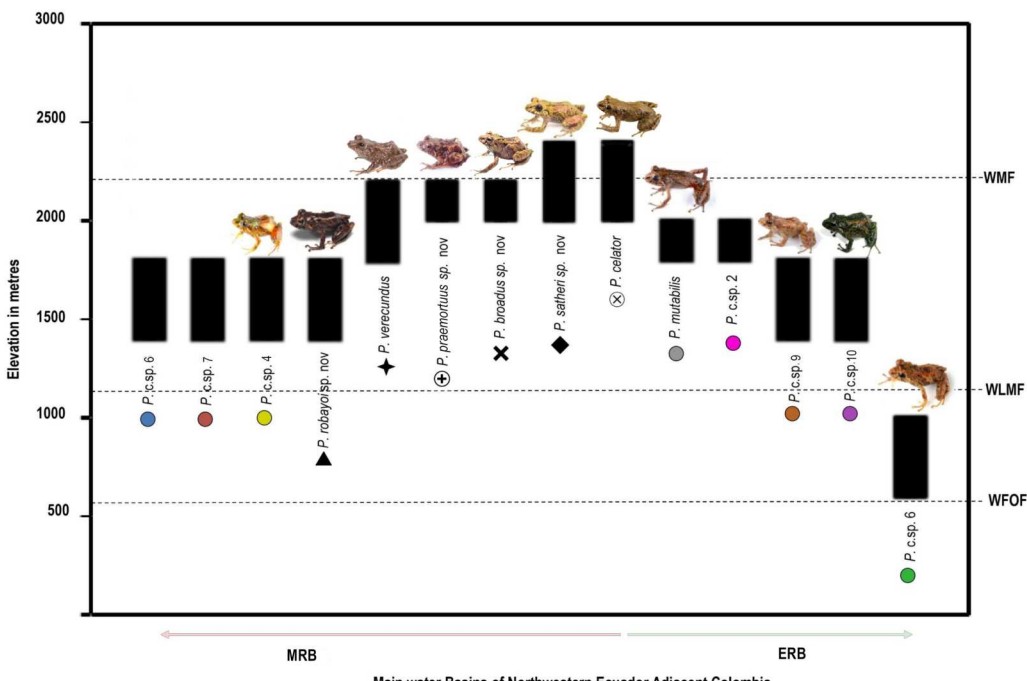

**Figure 16 Distribution and altitudinal range of *Pristimantis celator* clade.** Symbols referred to in the legend of Fig. 3. MRB, Mira River Basin; ERB, Esmeraldas River Basin. WMF, Western Montane Forest; WLMF, Western Low Montane Forest; WFOF, Western Foothill Forest Illustration by Mario H. Yánez-Muñoz.

the Esmeraldas River basin of Ecuador (Fig. 16). This distribution highlights clear allopatry within the clade, influenced by the Esmeraldas and Mira river watersheds, which have served as key factors in vertebrate divergence with sympatric species in similar altitudinal bands, approximately at 2,000 m (*Hutter, Guayasamin & Wiens, 2013*; *Mendoza et al., 2015*; *Antonelli et al., 2018*; *Perrigo, Hoorn & Antonelli, 2019*; *Franco-Mena et al., 2023*).

Re-examination and re-identification of *Pristimantis verecundus* Lynch & Burrowes type material, confirm the continuity in distribution of *P. verecundus* sensu stricto, with *P. satheri* sp. nov., and *P. robayoi* sp. nov. along the Mira River basin and its tributary micro-basins such San Juan River (Colombia-Ecuador border) and the Guiza River (La Planada Reserve, Colombia).

The presence of the Cerro Golondrinas mountain system within the altitudinal gradient of the Mira River basin, is a key element in the diversification of the *Pristimantis celator* clade, wide altitudinal differences, with sharp rocky walls and deep humid mountain valleys, promoting distinct environments, climate and subsequently speciation in our studied clade, as in other biological groups like orchids (*Baquero & Montero, 2020*; (*Monteros, Baquero & Vieira-Uribe, 2021*; *Monteros, Restrepo & Baquero, 2022*). The mountainous landscapes of the Mira River basin act as a vicariance barrier isolating sister species such as *Pristimantis praemortuus* sp. nov. and *Pristimantis broaddus* sp. nov., which exhibit genetic divergences exceeding 3.22% on its western and eastern slopes populations of the species, respectively.

Moreover, species on the western side of Cerro Golondrinas show patterns of altitudinal parapatry, with high variation in genetic distances, 10% and 11% between sisters species, showing a clear clinal pattern on the *Pristimantis celator* clade.

**Conservation Status.** We consider that the four newly described species have highly restricted ranges, closely tied to forested habitats, with distributions not exceeding 150 km$^2$. Although *Pristimantis verecundus* is currently listed as Vulnerable (VU B1a ($\leq$10) b (iii); *Ortega-Andrade et al., 2021*), our findings suggest an even more restricted range of less than 50 km$^2$.

All species in this complex were found outside Ecuador's National System of Protected Areas (SNAP), with distributions confined to private reserves and community-led conservation efforts, such as the Dracula Reserve, Cerro Golondrinas Protected Forest, and the ACUS Chinambí Conservation and Sustainable Use Area.

In Colombia, *Pristimantis verecundus*, *P. satheri* sp. nov., and *P. robayoi* sp. nov. persist in La Planada Reserve. The habitats of these species overlap with areas subject to mining exploration and exploitation (*Roy et al., 2018*; *Ortega-Andrade et al., 2021*; *Yánez-Muñoz et al., 2020*; *Yánez-Muñoz et al., 2024*). Our fieldwork has revealed that these new species, along with *Pristimantis verecundus*, exhibit low relative abundances, with fewer than ten individuals per species collected during each of the nine expeditions conducted in the study area. In contrast, other more abundant species of *Pristimantis*, such as *P. apiculatus*, are three times as common. Given these factors and the ongoing threats to their habitats, these species may warrant classification in a higher extinction risk category (*e.g.*, VU, EN, CR). Until further ecological and population data are available, we suggest assigning the new species as Data Deficient (DD) and re-evaluating the status of *Pristimantis verecundus*.

## CONCLUSIONS

This study proposes a redefinition of the *Pristimantis verecundus* clade as the *Pristimantis celator* clade, emphasizing the significance of utilizing the oldest available name, *Pristimantis celator*, for this group. This reorganization is consistent with phylogenetic evidence and clarifies the taxonomic status of species within the clade, which is now placed within the broader *Pristimantis myersi* species group under the subgenus *Trachyphrynus*.

The phylogenetic relationships and species delimitation of this study have identified and formally described four new species within the *Pristimantis celator* clade. These descriptions expand the known diversity within the clade from 10 to 14 species, underscoring the clade's cryptic diversity and the importance of thorough phylogenetic analyses for species delimitation.

Although the presence of a polytomy among three well-supported clades within the *Pristimantis myersi* species group, the study offers substantial evidence in support of the clade structure, particularly within the *Pristimantis celator* clade. The polytomy, likely influenced by incomplete genetic data for some species, does not detract from the evidence for the distinctiveness and evolutionary relationships among these groups.

The highly diversity observed within the cryptic *Pristimantis celator* clade can be attributed to the unique orogenic processes and the intricate geological and climatic

conditions of the extreme northwestern foothills of the Andes in Ecuador and southern Colombia, along with their associated watersheds. This underscores the clade's endemic nature and highlights the critical need for targeted conservation efforts. The study's findings enhance our understanding of the biogeographic patterns that influence species diversity in these regions, emphasizing the importance of preserving these habitats.

In the Mira Mataje landscape, at least three candidate species are distinct at the molecular level and remain undescribed. Further analysis with additional material is needed to understand the species boundaries and morphological variation within the montane ecosystems of the basin.

## ACKNOWLEDGEMENTS

We thank Paul A. Baker (Duke University) and Jorge Gómez (Beyond One & SDSN Andes) for their support, without which this work would not have been possible. We thank Sandra Galeano and Christian Venegas-Valencia from the Instituto de Investigación de Recursos Biológicos Alexander von Humboldt, for access to the type collections and for sending important high-resolution photographic material. We thank the collaboration during field work and collections by: Mateo A. Vega-Yánez, Daniel Padilla, Klever Jiménez and Rafael Mena (San Jacinto de Chinambí and Maldonado in 2019); Carlos Castro-Muñoz, Hugo Canchagua and Joseu Canchagua (Cabañas El Pailón, Bosque Protector Golondrinas in 2023); Jaqui Curay, Rocío Manovandas, Rubí García, Jorge Brito, Glenda Pozo, Jordi Salazar, Fausto Recalde, Gabriela Puetate, Sara Chingal, Callie Broaddus, Pearson McGovern, Carlos Ríos, Ross Mayner, Jaime Culebras, Natalia Espinoza, Daniel Valencia, Marco Monteros, Andy Better, Mauricio Herrera-Madrid. To Justin Sather, who motivated much of this research and was the trigger to deepen our investigation. To Callie Broaddus for all her affection in the pleasant and unforgettable field days and her wonderful photographic work in these last 6 years. To Javier "Capitán" Robayo for all the confidence and above all for the fun motivations during the field days. To the anonymous reviewers for their valuable contributions to the manuscript. Thanks for their valuable photographic contributions to: Santiago R. Ron (BioWeb-QCAZ), Jaime Culebras (Wildelife Tours), Mateo Vega-Yánez (INABIO), Christian Venegas-Valencias and Sandra Galeano (IvH). A special thanks from MYM to Mauro, Alejandra, Joaquín and Julieta, for their constant patience and love during the field days. The work of Mario H. Yánez-Muñoz and Juan Pablo Reyes-Puig is part of the research program Diversidad de Pequeños Vertebrados de Ecuador.

## APPENDIX 1. SPECIMENS EXAMINED

*Pristimantis broaddus* (nine): Ecuador: Carchi (nine): Cabañas El Pailón, Bosque Protector Golondrinas DHMECN 19037 (Holotype), DHMECN 19028–19029, DHMECN 19031–36. *Pristimantis mutabilis* (five): Ecuador: Imbabura (five): Lita, DHMECN 11755; Concesión Llumiragua DHMECN 13279; Concesión Cascabel ENSA, DHMECN 14796; Reserva Manduriacu DHMECN 16433; Reserva comunitaria Junín DHMECN

19479. *Pristimantis praemortuus* (nine): Ecuador: Carchi (nine): Los Olivos-Bloque 20, Reserva Dracula DHMECN 19592 (Holotype), DHMECN 19535, DHMECN 19546–47, DHMECN 19557, DHMECN 19570, DHMECN 19577, DHMECN 19579; San Jacinto de Chinambí, DHMECN 14887. *Pristimantis robayoi* (14): Ecuador: Carchi (13) Base del Golondrinas, Reserva Dracula, DHMECN 17894 (Holotype), DHMECN 16567, DHMECN 16573, DHMECN 16575, DHMECN 19429; San Jacinto de Chinambí DHMECN 14884, DHMECN14886; Río Cumbe, Maldonado DHMECN14947–48, DHMECN 14950, DHMECN 14953, DHMECN14960; La Esperanza, Quinshull DHMECN 16150. COLOMBIA (one): Dpto. de Nariño (one): Reserva La Planada IAvH-AM-1493. *Pristimantis satheri* (seven). Ecuador: Carchi (six): San Jacinto de Chinambí, DHMECN 14858 (Holotype), DHMECN 14885; Bloque 18, Reserva Dracula, DHMECN 16578, DHMECN 16574; Los Olivos-Bloque 20, Reserva Dracula DHMECN 19538; Base del Golondrinas, Reserva Dracula, DHMECN 17903. COLOMBIA: Dpto. de Nariño (one): Reserva La Planada, IAvH-AM-1492. *Pristimantis verecundus*: Ecuador: Carchi (nine): Reserva Dracula, Sector Cerro Negro, DHMECN 15007, DHMECN 15188; Reserva Dracula, sector Base del Golondrinas, DHMECN 16568–69; Reserva Dracula, sector Gualpi DHMECN 12597, DHMECN 15189, DHMECN 19432–33; Colombia: Dpto. de Nariño (2): Reserva La Planada IAvH-AM-1834 (Holotype), IAvH-AM-1457 (Paratype). *Pristimantis* sp. 3.: Ecuador: Carchi (one): Reserva Dracula: Sector El Pailón DHMECN 14006. *Pristimantis* sp. 4.: Ecuador: Carchi (two): Reserva Dracula: Sector El Pailón DHMECN 13984, DHMECN16572. *Pristimantis* sp. 6.: Ecuador: Pichincha (15): Reserva Experimental La Favorita DHMECN 1928, DHMECN 2023; Curipogio DHMECN 1927; 5918, 7399–06, 7437–39.

## APPENDIX 2. KEY OF IDENTIFICATION FOR FROGS OF THE *PRISTIMANTIS CELATOR* CLADE

1a. Large digits, dorsolateral dermal ridges, or glandular folds, conical elongate or subconical tubercles on eyelid, forearm and heel…………………………………...…………………….............2

1b. Short digits, without glandular folds or dorsolateral dermal ridges, without tubercles on eyelid, forearm or heel.......................................................................................................................*P. celator*

2a. Digital pads of toes narrow, about the width of the digit or 1.5X the size of the digit…………………………………………….…………………………….....……....3

2b. Digital pads of toes expanded, wider than the digit, 2X or more the width of the digit ……….4

3a. Tympanic annulus weekly defined and lateral fringes on fingers are weekly defined………………………………………………………………………...…… *P. broadus*

3b. Tympanic annulus clearly defined, and lateral fringes strongly crenulated mainly in finger IV ……………………………………………………………………………….. *P. praemortuus*

4a. Dorsolateral folds formed by dermal ridges, with large conical tubercles on the upper eyelid and heel. ………………………………………………………………………………………5

4b. Dorsolateral glandular fold and tibial gland, with small subconical tubercles on eyelid and heel ………………………………………………………………………………..*P. satheri*

5a. Dorsolateral folds extending posteriorly beyond the sacrum.………………………*P.mutabilis*

5b. Dorsolateral folds short, only reaching the sacrum, at the anterior border of ilium……………………….…………………………………………....……*P. verecundus*

5c. Dorsolateral fold extending to the sacrum, with two distinct black subconical scapular tubercle..……………………………………………………………………....…*P. robayoi*

### Funding

The work of Mario H. Yánez-Muñoz, Miguel A. Urgiles-Merchán, Gabriela Lagla-Chimba, Carolina Reyes-Puig and Julio C. Carrión-Olmedo was supported by the John D. & Catherine T. MacArthur Foundation for the project ''Biodiversity Conservation in the Mira and Mataje Rivers' Basins''. The work of Carolina Reyes-Puig is supported by COCIBA grants (HUBI ID: 48, 12268), USFQ and Diego F. Cisneros-Heredia at the Museo de Zoología, USFQ provided facilities. We received institutional support from Francisco Prieto and Diego Inclan of INABIO, and they (in addition to Pablo Sánchez, Jorge Brito M., Cesar Garzón S. and Maurico Herrera Madrid) provided academic support. The work of Juan P. Reyes-Puig is supported by Orchid Conservation Alliance OCA and the Reserva and Verein Dracula Reserve at Ecuador (the ''Dracula' Rangers'' of Ecominga Foundation: Hector Yela, Milton Canticuz, Jeovany Guerra, Rolando Peña, Nilo Ortiz, David Yela also provided collaboration during field work and collections), Lou Jost and all Ecominga Team which also supported the data collection. The funders had no role in study design, data collection and analysis, decision to publish, or preparation of the manuscript.

### Grant Disclosures

The following grant information was disclosed by the authors:
John D. & Catherine T. MacArthur Foundation.
COCIBA grants: HUBI ID: 48, 12268.
USFQ and Diego F. Cisneros-Heredia at the Museo de Zoología: HUBI ID: 48, 12268.
Francisco Prieto and Diego Inclan of INABIO.
Orchid Conservation Alliance OCA and the Reserva and Verein Dracula Reserve at Ecuador.

### Competing Interests

The authors declare there are no competing interests.

### Author Contributions

- Mario H. Yánez-Muñoz conceived and designed the experiments, performed the experiments, analyzed the data, prepared figures and/or tables, authored or reviewed drafts of the article, and approved the final draft.
- Juan P. Reyes-Puig performed the experiments, analyzed the data, authored or reviewed drafts of the article, and approved the final draft.
- Carolina Reyes-Puig performed the experiments, analyzed the data, authored or reviewed drafts of the article, and approved the final draft.
- Gabriela Lagla-Chimba performed the experiments, analyzed the data, prepared figures and/or tables, authored or reviewed drafts of the article, and approved the final draft.

- Christian Paucar-Veintimilla performed the experiments, analyzed the data, prepared figures and/or tables, authored or reviewed drafts of the article, and approved the final draft.
- Miguel A. Urgiles-Merchán performed the experiments, analyzed the data, prepared figures and/or tables, authored or reviewed drafts of the article, and approved the final draft.
- Julio C. Carrión-Olmedo conceived and designed the experiments, performed the experiments, analyzed the data, prepared figures and/or tables, authored or reviewed drafts of the article, and approved the final draft.

### Animal Ethics

The following information was supplied relating to ethical approvals (i.e., approving body and any reference numbers):

This research was conducted under permits granted for access to genetic resources: MAE-DNB-CM-2016-0045, N° MAE-DNB-CM-2019-0120, and MAATE-ARSFC-2023-3346 issued by the Ministry of Environment, Water, and Ecological Transition of Ecuador.

### Data Availability

The sequences are available at GenBank: PQ189053–PQ189081.

### New Species Registration

The following information was supplied regarding the registration of a newly described species:

Publication LSID: urn:lsid:zoobank.org:pub:0E31ADC7-D760-427D-A7F6-5E6ADB7A21B5

*Pristimantis praemortuus* sp. nov.: urn:lsid:zoobank.org:act:0C3E9CF7-3D61-4440-8203-C3E4A5D582C0

*Pristimantis broaddus* sp. nov.: urn:lsid:zoobank.org:act:F060D100-C4EA-4882-BCFE-5EEBD8D9FFB0

*Pristimantis satheri* sp. nov.: urn:lsid:zoobank.org:act:D756EFAA-9D18-4564-AB7F-B5E1866054E0

*Pristimantis robayoi* sp. nov.: urn:lsid:zoobank.org:act:FC56EB54-455A-4FCD-B2AD-0CAE25B12B47

### Supplemental Information

Supplemental information for this article can be found online at http://dx.doi.org/10.7717/peerj.18680#supplemental-information.

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
