# Peer review of "High speciation in the cryptic Pristimantis celator clade (Anura: Strabomantidae) of the Mira river basin, Ecuador-Colombia"

_PeerJ, doi:10.7717/peerj.18680_

## Round 0.1 · original submission · Major Revisions

Dear authors, I ask you to carefully bring this manuscript into compliance with the journal standards. I would like to draw special attention to the inadmissibility of using artificial intelligence in the process of writing the article. One of the reviewers of your article has drawn attention to this, and I would like to ask you to rewrite those fragments (if such fragments really exist in the manuscript) for which artificial intelligence was used.

Rounding of numbers in tables should be standardized within each of the characteristics.

In Table 2, one of the methods of multiple sample comparisons (for example, Tukey's test) should be correctly applied.

All fonts in Figure 1 should be easily readable, not very small.

·

Basic reporting

I could not locate the supplementary figure 7 mentioned in the manuscript. There are also inconsistencies in labeling the GenBank table and identification key that should be revised.

My other comments can be found throughout the text in the pdf file.

Experimental design

no comment

Validity of the findings

This is an important contribution that clarifies the identity of several lineages of Pristimantis, both known and new. The use of molecular tools, along with congruence to the extensive morphological data gathered from a significant number of specimens, supports the taxonomic decisions made. Overall, I find the manuscript well-written and detailed; the analyses conducted are appropriate, and the descriptions and comparisons are very thorough.

However, I have a main concern about species with “clear morphological differences” while being classified as cryptic. This discrepancy raises questions about the definition of "cryptic" as used in the context of this study. I recommend that the authors clarify their use of the term "cryptic".
In line with this, I noticed the use of the term "cryptic speciation". While this phrase has gained some traction in the literature, I find it conceptually imprecise. Speciation is a process, and processes themselves cannot be cryptic or non-cryptic. The term "cryptic" refers to the resulting species, not to the process of speciation itself.

Additional comments

I look forward to seeing it published after minor revisions are made to address the suggestions.

Congratulations to the authors!

·

Basic reporting

Overall the structure of the article is straightforward and conforms to what would be expected of a taxonomic revisionary study of Pristimantis species. New sequences and measurement data are reported. The most relevant existing literature studies are cited.

The one piece that is missing from the main text is an account of verecundus. It is stated that there is one in the supplementary files but no such text was available in the review materials. Ideally a diagnosis of verecundus would be included in the main text.

In reporting the results, the authors number some undescribed lineages that were previously numbered in a published study by Franco-Mena et al. It would be ideal if the same numbering was maintained in order to facilitate comparisons between these studies.

There are some typographical edits that should be addressed before publication to ensure uniformity of spelling, punctuation, capitalization, and usage. I've identified some specific examples below but recommend checking throughout for similar patterns.

1. italicization of species names missing: lines 35, 1078
2. "sp." or "complex" italics should be removed: lines 456, 634, 637
3. hyphen should be replaced by dash for numbered ranges: line 671 for example
4. remove double period: lines 220, 274, 780
5. inconsistent capitalization of "Finger", color terms, and "River": see for example lines 551, 552, 563, 600
6. inconsistent references to figures: see for example lines 749, 750
7. missing spaces or periods in "sp. nov.", "sp. 2", etc.: see for example lines 222, 235, 346, 507
8. inconsistent usage of "sp." vs "csp.": see for example line 260
9. misspellings of "redescription" (43), "morphometric" (98), "in" (320), "oblique" (377, 540), "white" (563), "nares" (705), "thenar" (721, 890), "juveniles" (807), "weakly" (870), "conical" (873), "finely" (885), "flesh" (927)
10. descriptive words or phrases like short, round, dorsal/back, etc. are repeated: see for example lines 317, 530, 543, 794, 869
11. formatting of the color numbering system is inconsistent: see lines 910-915
12. ensure spaces are present before and after "+" sign and absent before punctuation: see lines 235, 488, 493, 495
13. extra word fragments should be deleted: "[2]" (669), "lor" (718), "es" (887), "al" (889)
14. incomplete sentences: lines 238, 562
15. missing words: after "to" (330), after "positioned" (540)
16. inconsistent or incorrect table/figure references: lines 361, 527, 984. Note that the review manuscript did not include a supplementary table 2 or supplementary material 2.
17. "ICN" acronym on line 965 is missing context

Experimental design

The experimental design included both molecular and external morphological data collection sufficient to diagnose new species. My primary question in reading the study is why there were four species described but seven further lineages where maintained as candidate species. I'd recommend including a brief description in the methods of how it was determined which lineages had sufficient evidence to permit their formal description as new species.

In the methods, there are a few places where additional information or clarification would be useful.
1. On line 148, "pheotypic plasticity" is used. This term has a specific meaning in quantitative genetics that doesn’t fit this context. It seems like this is meant to state that the authors sought to minimize measurement changes due to effects of preservation. Also, it’s unclear whether this is stating that characters were scored 12 hours after collection or 12 hours after preservation.
2. On line 182, it is stated that 67 new sequences were generated but the actual number is 29 (some of which span multiple genes).
3. For the sequences used in the study, accession numbers of the new ones were provided, and Fig. 1A labels include some others, but that leaves several hundred published sequences that were used but do not have their accession numbers reported anywhere.
4. On line 193 the partitioning strategy should be described.


In the results, line 208 mentions eight partitions but only five are given, and there is no mention of how tRNAs were treated in the partitioning scheme. On line 213 and in the tree figures it is not stated which branch support measurement comes first and which comes second.

Validity of the findings

1. The background statement of polyphyly (line 78) is not true. The phylogeny in Franco-Mena et al shows a monophyletic verecundus, and an expanded definition that includes the undescribed forms within verecundus would also be monophyletic.

2. In deciding to use "celator clade" in place of "verecundus clade" (lines 224-226, 1078-1082), it should be acknowledged that “clade” isn’t a rank governed by the code, so the rules of priority don’t strictly apply. Thus, while using an older name is a reasonable preference it isn't a strict requirement to make this change.

3. In the diagnoses (lines 315, 482, 668, 824) tympanum diameters are reported as precise values. Surely there is individual variation within the series.

4. Also in the diagnoses, "males unknown" is given as a character twice (lines 321, 488). This is obviously not a diagnostic character and is contradicted by the presence of males in the type series for both species.

5. In the descriptions (lines 369, 533, 707, 872), percentages are reported but it is unclear what these percentages are.

6. On line 713, the description of tongue length and breadth is contradictory.

Additional comments

The way that the new species descriptions are described in the abstract is somewhat ambiguous. If reading only the abstract, it might be concluded that there are six species being described here (4 newly discovered ones, and 2 others split from verecundus), rather than the actual value of 4.

---

## Round 0.2 · Minor Revisions

Dear authors, I ask you to carefully check the manuscript, correct the minor technical flaws and send me the final version for acceptance for publication.

·

Basic reporting

This version is much clearer, and all previous issues have been addressed.
I have only a few minor suggestions:
In the P. praemortus account, there is an extra space on line 425. Additionally, the statement "Tympanic annulus..." starts a new sentence, whereas other traits are listed within the same sentence. Consider making this consistent.

In the P. broaddus section, "sp.nov" and "sp. nov" are used interchangeably; it would be best to standardize this. On line 619, "tympanic annulus" is duplicated, and on line 621, there is no space between "P. broaddus sp.nov." and the parenthesis ("fingers").

In the robayoi section, on line 1009, there is an uppercase "T" in "Tympanum" before a semicolon (";"). Furthermore, "Fingers" starts a new sentence here, but "tympanum" does not. Again, consider maintaining consistency in how these traits are listed.

I suggest the authors carefully review and revise the text for consistency in formatting and presentation throughout the manuscript.

Experimental design

no comment

Validity of the findings

no comment

Additional comments

After addressing these minor comments, I consider this work ready for publication.

---

## Round 0.3 · accepted · Accept

Dear authors, I congratulate you on the acceptance of this article for publication. I believe that it will generate considerable interest among readers. The material in the article is very informative and well-written. I wish you further success in your scientific research!